# Removal of extracellular human amyloid beta aggregates by extracellular proteases in *C. elegans*

Elisabeth Jongsma[1], Anita Goyala[1], José Maria Mateos[2], Collin Yvès Ewald[1]*

[1]Laboratory of Extracellular Matrix Regeneration, Institute of Translational Medicine, Department of Health Sciences and Technology, ETH Zürich, Schwerzenbach, Switzerland; [2]Center for Microscopy and Image Analysis, University of Zurich, Zurich, Switzerland

**Abstract** The amyloid beta (Aβ) plaques found in Alzheimer's disease (AD) patients' brains contain collagens and are embedded extracellularly. Several collagens have been proposed to influence Aβ aggregate formation, yet their role in clearance is unknown. To investigate the potential role of collagens in forming and clearance of extracellular aggregates in vivo, we created a transgenic *Caenorhabditis elegans* strain that expresses and secretes human $Aβ_{1-42}$. This secreted Aβ forms aggregates in two distinct places within the extracellular matrix. In a screen for extracellular human Aβ aggregation regulators, we identified different collagens to ameliorate or potentiate Aβ aggregation. We show that a disintegrin and metalloprotease a disintegrin and metalloprotease 2 (ADM-2), an ortholog of ADAM9, reduces the load of extracellular Aβ aggregates. ADM-2 is required and sufficient to remove the extracellular Aβ aggregates. Thus, we provide in vivo evidence of collagens essential for aggregate formation and metalloprotease participating in extracellular Aβ aggregate removal.

*For correspondence:
collin-ewald@ethz.ch

**Competing interest:** The authors declare that no competing interests exist.

## Editor's evaluation

It is of significance to generate a novel transgenic *C. elegans* model with inducible expression and secretion of human GFP-tagged human Aβ1-42 for the neurodegenerative disease field. This paper is interesting to neuroscientists who work on protein aggregation and neurodegenerative diseases. The key claims of the manuscript are well supported by the revised convincing data revealing that the metalloprotease ADM-2 modulates ECM and assists in the removal of extracellular Aβ aggregates.

## Introduction

Alzheimer's disease (AD) currently affects >1 in 9 people above 65 years of age (11.2%) in the USA and is the seventh cause of death worldwide (*Alzheimer's Association, 2022*; *Mortality and global health estimates, 2019*). A hallmark of AD is the extracellular aggregation of amyloid beta (Aβ). Over the past decades, much knowledge has been gained on the production and removal of Aβ. Several mechanisms are involved in clearing Aβ from the brain, including enzymatic and non-enzymatic pathways. The non-enzymatic pathways include the continuous flow of the interstitial fluid into the cerebrospinal fluid followed by interstitial fluid drainage, phagocytosis by microglia or astrocytes, and receptor-mediated transport across the blood-brain barrier (*Elbert et al., 2022*; *Sagare et al., 2007*; *Tajbakhsh et al., 2021*; *Zhao et al., 2015*). The enzymatic pathway involves several proteases, including matrix metalloproteinases (MMP), neprilysin, insulin-degrading enzymes, and glutamate carboxypeptidase

(*Nalivaeva et al., 2012*). The extracellular protein heparan sulfate proteoglycans (HSPGs) can block the clearance of Aβ. HSPGs are often found in Aβ depositions where they might block enzymatic degradation (*Gupta-Bansal et al., 1995*; *Su et al., 1992*; *van Horssen et al., 2002*). Moreover, while most HSPGs promote the uptake of Aβ through lipid rafts, uptake of Aβ through clathrin-mediated endocytosis is blocked when the HSPG (SDC3) binds to Aβ (*Letoha et al., 2019*).

One of the least understood observations is the consistent co-aggregation of specific collagens with Aβ plaques. Interestingly, the compaction of Aβ into plaques can be influenced by the expression of collagenous amyloid plaque components (CLACs) (*Hashimoto et al., 2020*). CLAC is a collagen type XXV a1 chain (COL25A1) cleavage product. COL25A1 overexpression can have detrimental effects in mice (*Tong et al., 2010*). However, human genetic studies suggest a more complex interplay where certain single nucleotide polymorphisms in COL25A1 are associated with AD and others are, in contrast, associated with health effects in the elderly (*Erikson et al., 2016*; *Forsell et al., 2010*). Curiously, several other collagens have been found to have a protective role. Colocalizing with vascular amyloid at the basal lamina is collagen XVIII, an HSPG that reduces disease symptoms (*van Horssen et al., 2002*). Collagen VI was found at the extracellular matrix (ECM) and the basal lamina and can block the interaction between neurons and oligomers and help protect against neurotoxicity (*Cheng et al., 2009*; *Ma et al., 2020*). Furthermore, the basement lamina collagen IV was 55% upregulated in cerebral vessels when comparing AD to healthy subjects (*Cheng et al., 2009*; *Farkas et al., 2000*; *Kalaria and Pax, 1995*; *Nguyen et al., 2021*). A positive correlation was reported between disease progression (Braak stage) and collagen IV deposition. With each group: control (Braak stage <2), preclinical (Braak stage >2, <4), and AD patients (Braak stage >4), an increase in collagen IV deposition was found per mm2 gray matter (*Lepelletier et al., 2017*). Collagen IV was shown to bind the amyloid precursor protein (APP), prevent Aβ fibril formation, and even disrupt preformed Aβ fibrils (*Kiuchi et al., 2002a*; *Kiuchi et al., 2002b*; *Narindrasorasak et al., 1995*). Based on these collected studies, we hypothesized that the ECM may be more than a passive bystander and that its components hold the potential to influence disease progression.

To address this hypothesis, we set out to explore the mechanisms by which ECM components influence Aβ aggregate formation and clearance in vivo. However, a model monitoring this in vivo and non-invasively was missing. Therefore, we generated a novel transgenic *Caenorhabditis elegans* strain with inducible expression and secretion of human Aβ$_{1-42}$ tagged with super-folder GFP (sfGFP::Aβ). Furthermore, *C. elegans* is a suitable model to address this question. Several Alzheimer-related pathways are highly conserved between humans and *C. elegans* (*Apostolakou et al., 2021*; *Ewald and Li, 2010*). Moreover, the ECM components associated with AD have orthologs in *C. elegans*. The *C. elegans* EMB-9 and LET-2 are collagen type IV, CLE-1 is collagen type XVIII, and COL-99 is collagen type XXV (*Teuscher et al., 2019a*). While EMB-9 and LET-2 localize to the basal lamina, CLE-1 and COL-99 localize to neurons.

Here, we show that upon induction, secreted sfGFP::Aβ is initially cleared by the excretory system, the gut, and the coelomocytes. However, Aβ is retained past 24 hr and forms non-mobile structures in the ECM. We identified collagens that can completely suppress Aβ aggregate formation. Moreover, we find modulators of the ECM, metalloproteases, to assist in the removal of extracellular Aβ aggregates. We demonstrate that one of these metalloproteases, a disintegrin and metalloprotease 2 (ADM-2), is essential to remove Aβ aggregates. Taken together, this suggests that ECM composition is critical to allow Aβ aggregate formation, while dynamic regulation of the ECM through metalloproteases is key in Aβ aggregate clearance.

## Results

### Generating an in vivo model for extracellular Aβ aggregates

An obstacle to studying the interaction of the ECM with Aβ aggregation and clearance is the lack of an in vivo model. Previous human Aβ expressing *C. elegans* strains failed to secrete Aβ and model intracellular Aβ toxicity (*Ewald and Li, 2012*; *Link, 1995*). Therefore, we designed a genetic construct that secretes Aβ tagged with GFP (*Figure 1A and B*). Expression of this construct was induced by heat shock under the control of the *hsp-16.2* promoter that drives expression in many tissues but predominantly in neurons and hypodermis (*Bacaj and Shaham, 2007*). Furthermore, the construct has a longer 3' UTR targeting its mRNA for non-sense-mediated degradation to prevent the leakage

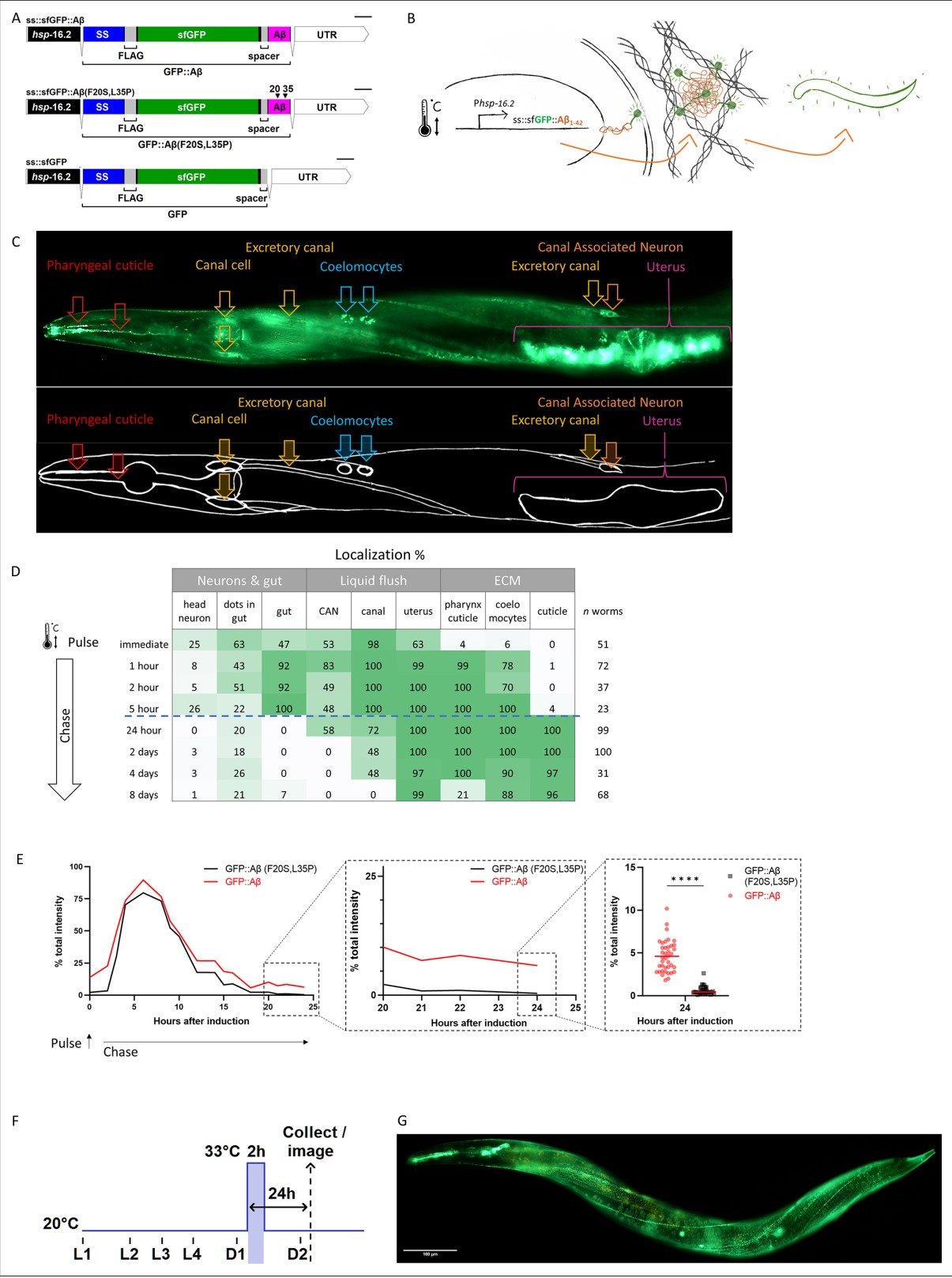

**Figure 1.** Expression of secreted human amyloid beta (Aβ) tagged with super-folder GFP (sfGFP). (**A**) Genetic constructs used to generate transgenic *C. elegans* strains. SS indicates secretion sequence, UTR = untranslated region. (**B**) Hypothetical model of induction, expression, and secretion of sfGFP::Aβ in *C. elegans*. After a single heat-shock induction, sfGFP::Aβ is secreted and localizes to different tissues over time. (**C**) Localization of sfGFP::Aβ to different tissues in *C. elegans*. The localization to the excretory canal and the coelomocytes confirm that the sfGFP::Aβ is secreted. (**D**)

*Figure 1 continued on next page*

*Figure 1 continued*

Percentage of tissue type with sfGFP::Aβ over time. After production and secretion, most of the produced sfGFP::Aβ was flushed out, but some were retained at the cuticle up to 8 days after the induction event. (**E**) Clearance of sfGFP::Aβ is significantly slowed >18 hr after induction compared to non-aggregating control sfGFP::Aβ(F20S, L35P). Data represented is the average from three independent repeats combined; repeats are shown in *Figure 1—figure supplement 1E*. (**E**) sub III image is from one of the repeats, unpaired, two-tailed t-test. ****: p<0.0001. (**F**) Representation of methods regarding the time of heat-shock induction of expression and imaging or sample collection 24 hr after induction. (**G**) Representative image of the transgenic line LSD2104 and localization of secreted sfGFP::Aβ.

The online version of this article includes the following source data and figure supplement(s) for figure 1:

**Figure supplement 1.** Validation of human amyloid beta (Aβ) present in transgenic sfGFP::Aβ *C. elegans*.

**Figure supplement 1—source data 1.** Full raw unedited western blots and uncropped and labeled western blots.

**Figure supplement 2.** Time course of secreted amyloid beta.

**Figure supplement 3.** Quantification of the secreted amyloid beta time course.

of the *hsp-16.2* promoter (*Ewald et al., 2016*). This allowed us to separate events scaled in time, for example, deposition versus removal of Aβ. We used sfGFP because it is more stable in the extracellular space than classical GFP (*Pédelacq et al., 2006*). A spacer sequence was placed between the sfGFP and the Aβ to allow the comparably smaller-sized Aβ to move and interact freely to form aggregates (*Figure 1A and B*). The full length of the Aβ$_{1-42}$ peptide is essential for its aggregation (*McColl et al., 2012*). In our construct, Aβ is preceded by the sfGFP and spacer sequence, which prevents the truncation of the first few amino acids observed in many previous *C. elegans* Aβ models (*McColl et al., 2009*). As controls, we generated two constructs: one containing a non-aggregating version of Aβ$_{1-42}$ (secreted sfGFP::Aβ(F20S, L35P)) (*Wurth et al., 2002*), and the other control is the secreted sfGFP without Aβ fragment (*Figure 1A*).

## Dynamic turnover of secreted Aβ

After a single heat-shock induction of sfGFP::Aβ (i.e., single pulse-chase), sfGFP::Aβ was expressed in many tissues and secreted into the extracellular space surrounding different tissues (*Figure 1C*). This Aβ localization changed over time. Initially, Aβ localized to neurons in the nerve ring, the gut, the canal cells, the excretory canal, the canal-associated neurons, and the uterus (*Figure 1C and D*). During the first 24 hr, most of the sfGFP::Aβ appeared diffuse. After 24 hr, we observed bright puncta localized to the pharyngeal cuticle, the coelomocytes, and the cuticle (*Figure 1D*). By western blotting using GFP or Aβ antibodies, we confirmed that these sfGFP::Aβ fluorescent puncta contain the human Aβ (*Figure 1—figure supplement 1*). Remarkably, these sfGFP::Aβ fluorescent puncta remained for an exceptionally long time at the cuticle, coelomocytes, and the uterus, up to 8 days after induction (*Figure 1D*).

To determine whether this is due to the potential aggregation of Aβ, we followed sfGFP::Aβ intensity in a pulse-chase time course, including our two controls. After heat shock, all three strains showed similar time course trajectories for the induction (*Figure 1—figure supplement 2*). However, at the 24 hr time point where the sfGFP::Aβ was observed at the cuticle, the non-aggregating sfGFP::Aβ(F20S, L35P), and the sfGFP-only strains almost completely lost the GFP signal (*Figure 1E*, *Figure 1—figure supplements 2–3*). No localization to the cuticle was observed at this time point. Taken together, this suggests these sfGFP-linked Aβ were secreted and efficiently cleared from the extracellular spaces. At the same time, the retention at the cuticle after 24 hr was specific to Aβ$_{1-42}$ due to its ability to form aggregates. In search of mechanisms selectively affecting the aggregation-prone Aβ, all further interventions were assessed at 24 hr (*Figure 1F*). Hence, we have generated a transgenic strain expressing and secreting aggregation-prone sfGFP::Aβ that, after initial clearance, is retained at the cuticle (*Figure 1G*).

## sfGFP::Aβ forms 'flower' and 'moss' patterns in the extracellular space

To characterize the potential sfGFP::Aβ aggregates in the extracellular space, we compared the non-aggregating sfGFP::Aβ(F20S, L35P) with the aggregation-prone sfGFP::Aβ. At 16 hr post-induction, the non-aggregating sfGFP::Aβ(F20S, L35P) formed a striped pattern, equally distributed along the body, reminiscent of the striped pattern of furrows on the cuticle. This signal was relatively weak and uniform, similar to the GFP-only strain (*Figure 2A* [intensity enhanced], *Figure 2—figure supplement*

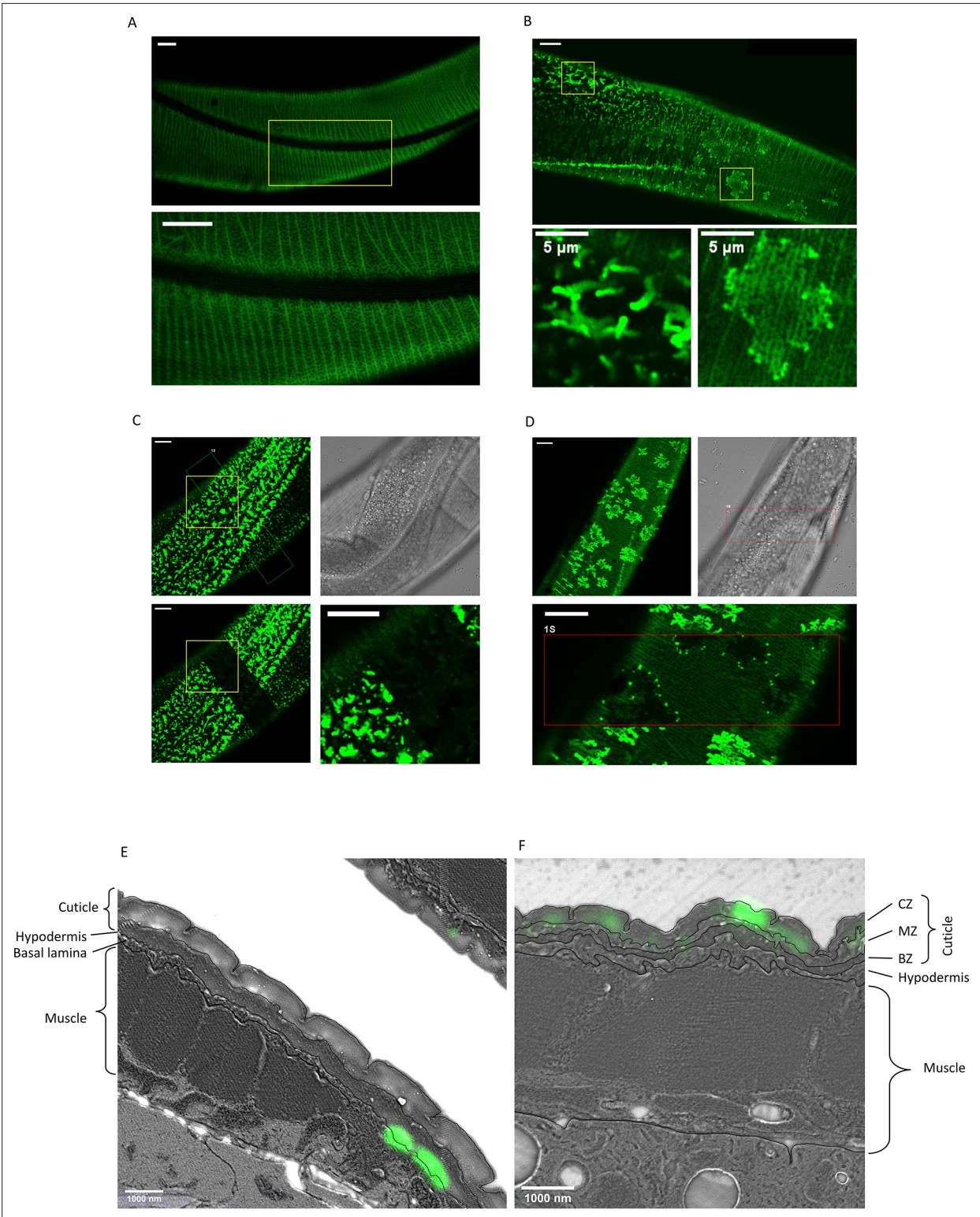

**Figure 2.** Secreted amyloid beta (Aβ) form aggregates at *C. elegans* extracellular matrix (ECM). (**A**) The control strain sfGFP::Aβ(F20S, L35P) shows localization near the cuticle. However, the signal is relatively weak and uniform. Note. Image intensity enhanced, taken 16 hr after induction. (**B**) Two types of bright patterns, dubbed 'moss' and 'flower', can be observed for sfGFP::Aβ near the cuticle, 24 hr past induction of expression. (**C, D**) Fluorescence recovery after photobleaching shows both the moss (**C**) and flower (**D**) structures are immobile. Time of imaging up to 4 hr after bleaching. (**E, F**) Correlative light electron microscopy revealed localization to ECM structures. (**E**) Localization of the 'moss' structures to basal lamina when there

*Figure 2 continued on next page*

*Figure 2 continued*
is no muscle underneath. (**F**) Localization of the 'flower' structures to the cuticle when there is muscle underneath. CZ: cortical zone of the cuticle, MZ: medial zone of the cuticle, BZ: basal zone of the cuticle. Scale bars are 10 µm unless otherwise indicated.

The online version of this article includes the following figure supplement(s) for figure 2:

**Figure supplement 1.** Flower and moss structures were not observed in the GFP-only strain.

**Figure supplement 2.** Location of sfGFP::Aβ apical to the hypodermis.

**Figure supplement 3.** Location of sfGFP::Aβ above the hypodermis.

**Figure supplement 4.** Location of sfGFP::Aβ in the cuticle.

*1*). By contrast, sfGFP::Aβ formed two types of bright structures, which we named 'flower' and 'moss' (*Figure 2B*). Interestingly, the 'flower' structures were exclusively found over the nematode's dorsal and ventral sides, where muscle tissue underlies the hypodermis. The 'moss' structures were consistently found on the left and right sides of the nematode, apical to the main hypodermal syncytium (hyp7). The localization of the aggregate type is invariant and correlates with the underlying tissue type. The flower and moss structures are not found for the non-aggregating sfGFP::Aβ(F20S, L35P) (*Figure 2A*) nor for GFP-only (*Figure 2—figure supplement 1*) but were unique to the wild-type sfGFP::Aβ strain.

## The 'flower' and 'moss' patterns are immobile aggregates

Next, to determine whether these flower and moss structures could be aggregates, we used fluorescence recovery after photobleaching (FRAP), a technique commonly used to determine if the tagged protein of interest is fixed in place. A small area within the image field was photobleached using a brief exposure to UV, quenching the GFP fluorescence. Free movement or transport of the fluorescently tagged protein should lead to an exchange of molecules between the bleached and unbleached area, resulting in a recovery of fluorescence in the bleached (dark) area over time. In the moss structure, no such recovery was observed (*Figure 2C*). Even when one-half of a single moss structure was photobleached, there was no recovery in the bleached area, indicating no movement or exchange of molecules even within the structure (*Figure 2C*). Upon photobleaching, the flower structures turn completely dark (*Figure 2D*). The surrounding, relatively low GFP intensity recovered rapidly (within seconds), but the flower structure remained dark. No recovery of signal inside the structure was observed; however, after 4 hr, some bright dots were observed on the edge of the flower structures, indicating potential growth or exchange of Aβ from outside the photobleached area. However, this was observed only at the border and not at the center of the flower structures (*Figure 2D*), suggesting confinement on the upper and lower part of this structure as it would be 'sandwiched' between something. For both the flower and moss structures, the absence of fluorescence recovery for up to 4 hr confirmed that these structures are immobile, suggesting that these structures are aggregates.

## Distinct aggregate patterns on the cuticular ECM

To determine where these structures localize, we used correlative light and electron microscopy (*Mateos et al., 2018*). This technique localizes the fluorescence signal in relation to the morphology of the tissue on the same thin sections (110 nm), providing a high X, Y, and Z resolution. The sfGFP::Aβ was observed to localize to two different parts of the cuticular ECM (*Figure 2E and F*, *Figure 2— figure supplements 2–4*). The localization apical to the hypodermis, underneath the collagen-dense cuticle, is representative of the 'moss' type aggregates since these are associated with the hypodermis (*Figure 2E*, *Figure 2—figure supplements 2–3*). By contrast, the localization to the cortical layer of the cuticle is representative of the 'flower' type aggregates since these are associated with underlying muscle tissue (*Figure 2F*, *Figure 2—figure supplement 4*). Taken together, we established a novel in vivo model of secreted human Aβ that forms aggregates in the ECM.

## Screening identified clearance mechanisms of Aβ aggregates in the ECM

To identify key molecular players in the formation and clearance of Aβ aggregates localized in the ECM, we designed a targeted RNA interference (RNAi) screen. We rationalized that examining four main categories of genes could elucidate a role for ECM molecules in Aβ aggregation. These categories

were: all known ECM genes and ECM remodeling genes (matrisome; n=719) (*Teuscher et al., 2019a*), genes involved in attaching the cell to the ECM and mechanosensation (n=255), *C. elegans* orthologs of genes associated with human AD (n=776) (*Vahdati Nia et al., 2017*) as well as genes protecting against neurodegenerative and age-related pathologies (n=631) (*Figure 3A*, *Supplementary file 1*).

To assess the effects of individual gene knockdown on aggregation in the ECM, an increase or decrease in sfGFP::Aβ signal was scored 24 hr post-induction (*Figure 3B*), the time point at which non-aggregating Aβ and soluble GFP were flushed out. We identified 176 from 2368 screened genes to increase or decrease sfGFP::Aβ intensity in vivo (*Figure 3A*; *Supplementary file 1*). To prioritize hits, we categorized these candidate genes into four groups (*Figure 3C*, *Supplementary file 2*). In category 1, we grouped hits expected to affect sfGFP::Aβ, such as genes involved in protein homeostasis, chaperones, and protein degradation (*Figure 3C*, *Supplementary file 2*). Reassuringly, we identified 71 orthologs of AD implicated genes (category 2, *Figure 3C*, *Supplementary file 2*), suggesting the conservation of key molecular players relevant to human disease. We also identified interesting hits with less established relationships to AD, such as vesicle transport and vesicle fusion, as well as members of the SNARE complex (category 3, *Figure 3C*, *Figure 3—figure supplement 1*, *Supplementary file 2*). When interrogating these hits with follow-up experiments, the change in sfGFP::Aβ aggregation could be explained by alternative mechanisms, such as impaired secretion or endosome recycling. Lastly, in category 4, we found that the knockdown of MMP increased Aβ levels, whereas the knockdown of their inhibitors (tissue inhibitor of metalloproteases [TIMP]) decreased the Aβ levels (*Figure 3C*, *Supplementary file 2*). Furthermore, the knockdown of several individual collagens either increased or decreased sfGFP::Aβ fluorescence, which reinforced the idea of a potentially active role of ECM components in Aβ aggregation and Aβ aggregate removal (*Figure 3C*, *Supplementary file 2*).

## Collagens implicated in Aβ aggregate formation

To determine the role of collagens in Aβ aggregation and Aβ aggregate removal, we examined flower and moss Aβ aggregates upon collagen knockdown. We found that the knockdown of cuticular collagen *col-2* or *col-79* resulted in more animals with flower Aβ aggregates (*Figure 4A*). By contrast, cuticular collagens *dpy-3* and *col-89* knockdown resulted in the complete absence of moss and flower Aβ aggregates, and *col-8*(RNAi) showed a marked reduction in Aβ aggregates (*Figure 4A*). As previously reported, DPY-3 is required for furrow formation of the cuticle, and animals lacking DPY-3 show disturbance of cuticular organization combined with a shortened and thicker body shape (*Sandhu et al., 2021*). DPY-3 is expressed during early developmental stages, while COL-8 and COL-89 are expressed during the last larval stage L4. The aggregates may require a particular ECM composition or specific collagens to form aggregates, or the knockdown of some collagens triggers overall ECM remodeling aiding the removal of aggregates. To separate these possibilities, we compared the knockdown of these collagens starting at different time points: RNAi beginning at the first larval stage (L1) to RNAi starting at the last larval stage (L4). We found that DPY-3 was required from early development for aggregates to form but not at later stages (*Figure 4B and C*). For COL-8 and COL-89, the effects of RNAi on intensity and aggregates were present when knocked down from the L4 stage (*Figure 4B and C*). Indeed, this suggests that the presence of these structural components of the ECM is either directly or indirectly required for Aβ aggregate formation.

To further establish collagen dynamics and ECM remodeling in Aβ aggregate load in the cuticle, we treated sfGFP::Aβ transgenic animals 24 hr prior to heat-shock induction with pharmacological inhibitors of either collagen synthesis or cross-linking and quantified sfGFP::Aβ levels 24 hr post-heat-shock induction when sfGFP::Aβ formed aggregates in the cuticle. Using collagen synthesis inhibitor 2,2'-bipyridine (BPY) that binds in the center in the prolyl 4-hydroxylase (*Hales and Beattie, 1993*), which blocks the addition of hydroxyproline to the collagen chain in the endoplasmic reticulum and thereby destabilizes collagen fibers, we observed lower overall sfGFP::Aβ fluorescence (*Figure 4—figure supplement 1A*). By contrast, collagen cross-linking inhibitor β-aminopropionitrile (BAPN), a lysyl oxidase inhibitor (*Ida et al., 2018*), resulted in higher overall sfGFP::Aβ fluorescence (*Figure 4—figure supplement 1B*). These data indicate that collagen biogenesis and collagen stability can affect Aβ aggregate levels.

Next, we went back to our screening hits to determine whether the four conserved collagens (collagen type IV [*let-2* and *emb-9*], collagen type XVIII [*cle-1*], and collagen type XXV [*col-99*]),

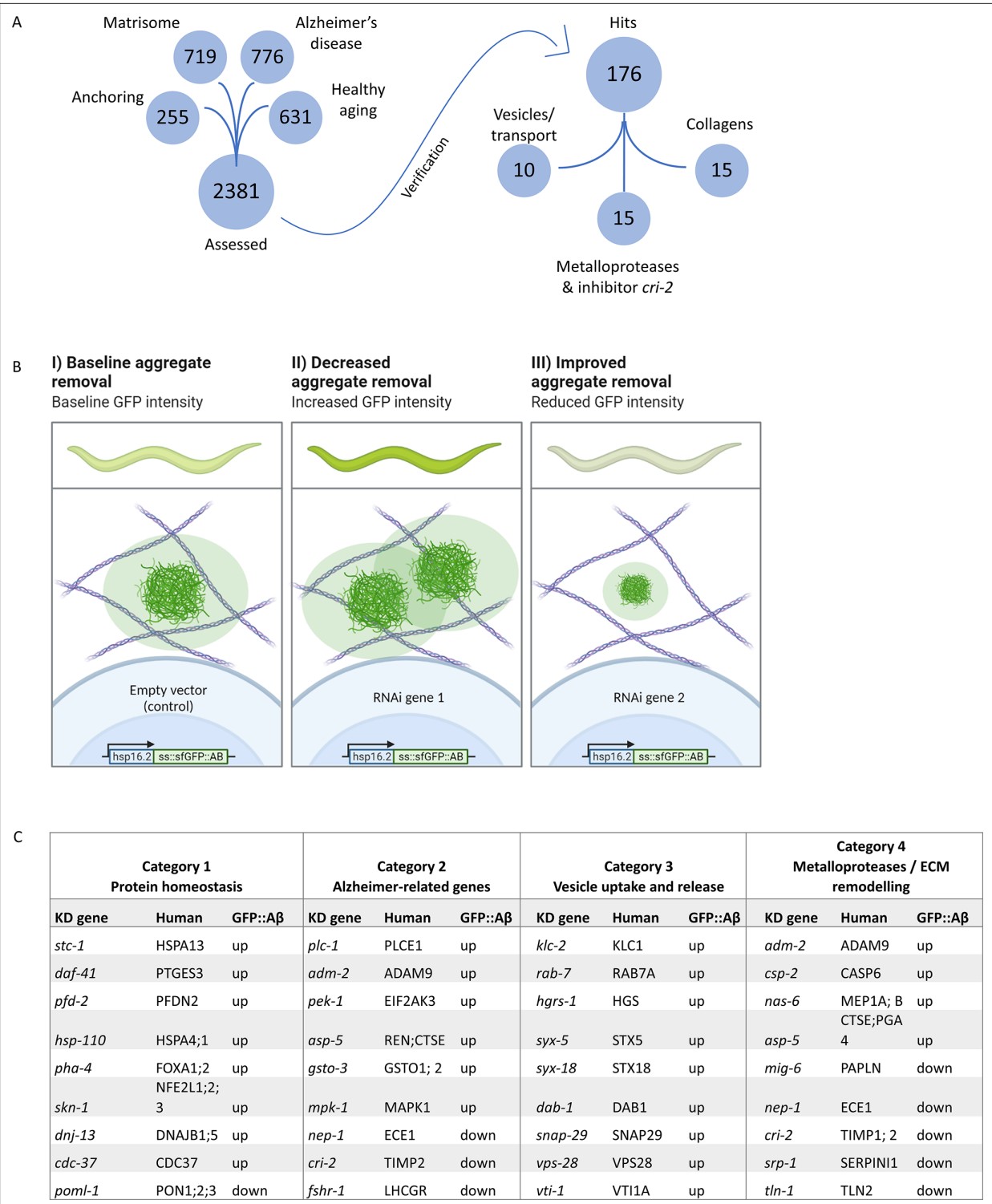

**Figure 3.** Strategy for RNA interference (RNAi) screening identified genes involved in the removal of extracellular sfGFP::Aβ. (**A**) Four RNAi libraries were designed based on their hypothesized potential to affect extracellular sfGFP::Aβ aggregation. The Matrisome library contains all extracellular *C. elegans* genes, the Anchoring library contains transmembrane genes, the Alzheimer's disease library is based on a meta-analysis of *C. elegans* orthologs of human GWAS, and the healthy aging library consists of genes that have a protective role against aging-related disease. Of the 2381 genes assessed, 176 genes were found to either increase or decrease sfGFP::Aβ load. (**B**) Expected fluorescence phenotypes of suppressor or enhancer genes. Grown on individual RNA clones from the L1 larval stage, populations of about 45 animals were assessed for an increase or decrease in GFP signal. An increase in signal would indicate decreased aggregate removal, while a decrease of GFP signal would imply improved aggregate removal upon

eLife Research article

Cell Biology | Neuroscience

knockdown of target gene expression. (**C**) Summary of categorization of hits shows relevant mechanisms to extracellular sfGFP::Aβ aggregate removal. Category 1 revealed screen hits in protein homeostasis, as expected when observing protein expression and turnover. Category 2 showed screen hits of orthologs of genes well known to be associated with Alzheimer's disease in humans. Category 3 revealed screen hits of genes involved in vesicle uptake and release, processes essential to secretion and removal of extracellular proteins such as sfGFP::Aβ. Category 4 showed the involvement of metalloproteases and an inhibitor of metalloproteases. Selected candidate hits are shown; the full table and raw RNAi score are available in **Supplementary file 2**.

**Figure supplement 1.** Vesicle and transport screening hits might prevent extracellular amyloid beta aggregation simply by blocking export.

which had previously been found to influence Aβ aggregation in mammals, would also have a functional role in our system. CLE-1 has been reported to be expressed in neurons and muscles and localizes predominantly around synapses and neuromuscular junctions (**Ackley et al., 2001**; **Heljasvaara et al., 2017**). Since RNAi is incompletely penetrant and less effective in neurons (**Asikainen et al., 2005**), we used a mutant for this neuronal-expressed collagen. We found that collagen type XVIII ortholog *cle-1(gk364)* mutants showed lower overall sfGFP::Aβ fluorescence and a mild reduction in moss and flower aggregates (**Figure 4D and E**). Since CLE-1 does not localize to the cuticle, this suggests indirect effects of *cle-1* collagen on Aβ aggregates. For collagen type XXV, *col-99* in *C. elegans*, RNAi showed no change in sfGFP::Aβ fluorescence intensity (**Figure 4F**). Neither *col-99(ok1204)* mutants nor COL-99 overexpression showed any consistent effect on sfGFP::Aβ fluorescence intensity (**Supplementary file 3**). For collagen type IV (*let-2* and *emb-9*), *emb-9* RNAi showed no changes in sfGFP::Aβ fluorescence intensity but showed a mild developmental delay (**Figure 4F**). Assessment using RNAi from the L4 stage again showed no influence on sfGFP::Aβ fluorescence intensity (**Supplementary file 1**). Furthermore, overexpression of EMB-9 did not noticeably change sfGFP::Aβ aggregation, nor did sfGFP::Aβ and EMB-9::mCherry colocalize (**Figure 4G and H**, **Figure 4—figure supplements 2–4**), which could explain why we observed no effect of collagen type IV on sfGFP::Aβ aggregation in this model. In summary, although we observed that cuticular collagens could influence Aβ aggregation and Aβ aggregate removal, the previously implicated orthologs might not directly affect sfGFP::Aβ aggregation in the cuticular ECM. Nevertheless, our data points toward collagen and ECM remodeling influencing Aβ aggregation and Aβ aggregate removal.

## TIMP and MMPs regulate Aβ removal

To define a role for collagen and ECM remodeling in the development and removal of Aβ aggregates at the cuticle, we used a broad-range inhibitor of metalloproteases, batimastat (BB94) (**Jacobsen et al., 2010**). Consistent with our RNAi screening hits on extracellular proteases, exposure to batimastat increased the sfGFP::Aβ load, represented by an increase in GFP intensity (**Figure 5A and B**), suggesting that metalloprotease activity is essential in removal.

Metalloprotease activity is controlled by TIMPs (**Nagase et al., 2006**). Interestingly, we also picked up *cri-2* in the screen, an ortholog of human TIMP (**Teuscher et al., 2019a**). Knockdown of *cri-2* resulted in reduced sfGFP::Aβ intensity, suggesting that when the inhibitor of metalloproteases is removed, the metalloproteases increase their activity and remove the sfGFP::Aβ. To validate this, we crossed in a genetic deletion, *cri-2(gk314),* and found lower levels of sfGFP::Aβ compared to a wild-type background at the 24 hr time point (**Figure 5C**). This reduction is not due to a difference in the initial expression of sfGFP::Aβ, as there was no difference between intensities in the *cri-2* mutant and wild-type backgrounds over the first 20 hr after induction (**Figure 5—figure supplement 1**). To identify the metalloproteases inhibited by CRI-2, we tested all MMPs identified in our screen in the *cri-2* mutant background (**Figure 5D and E**). From the nine metalloproteases tested, the knockdown of *adm-1, adm-2, adt-2,* and *mig-6* resulted in increased GFP intensity in both the wild-type and *cri-2* mutant backgrounds (**Figure 5D and E**). This suggests that when the inhibitor CRI-2 is absent, these four metalloproteases become more active and contribute to the removal of sfGFP::Aβ. Knockdown of *adm-2* outperformed the other metalloproteases as indicated by the higher intensity in the *cri-2* mutant background (**Figure 5E**), suggesting a more prominent role for ADM-2.

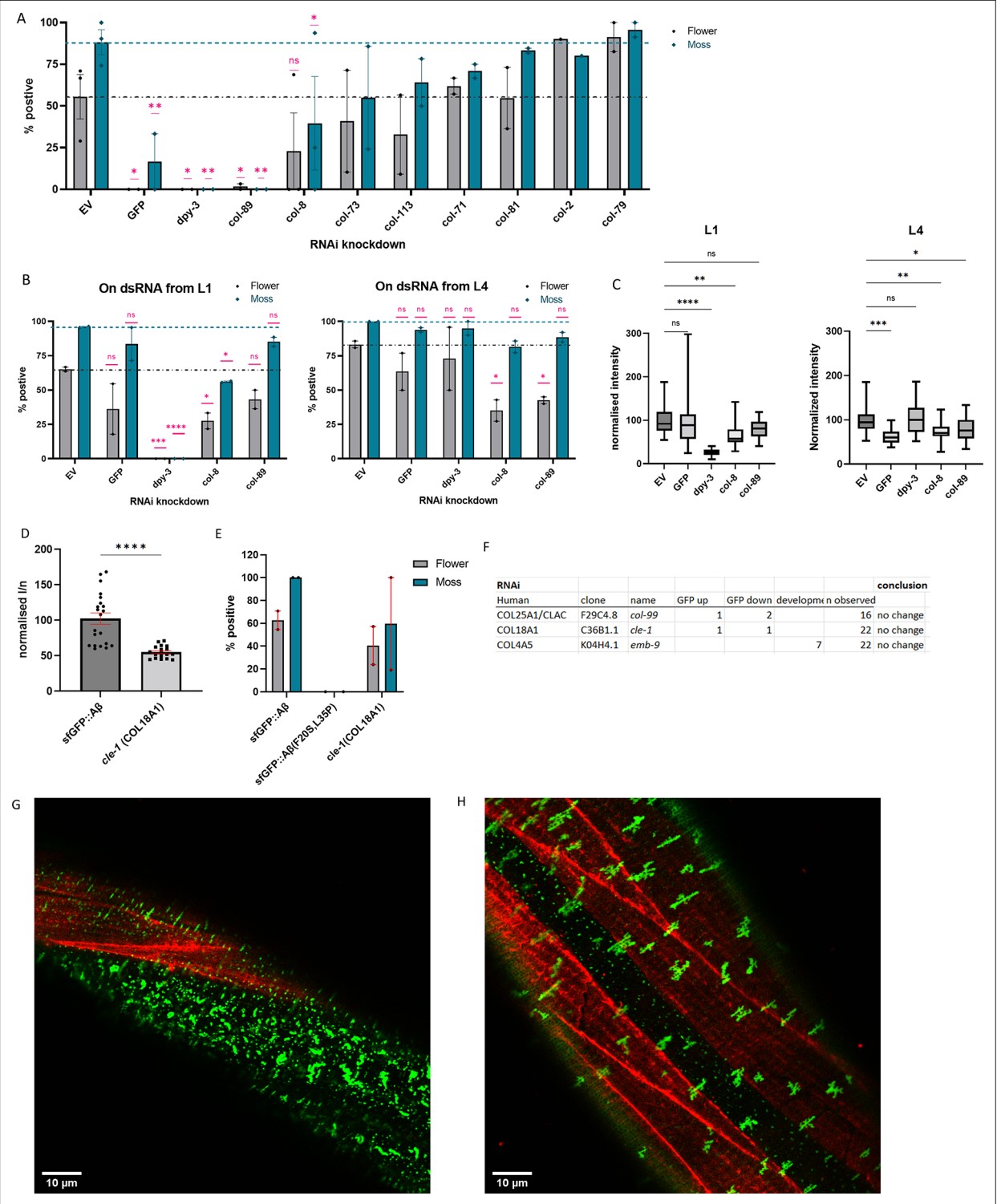

**Figure 4.** Collagens knockdown prevented or promoted extracellular sfGFP::Aβ aggregation. (**A**) Collagen knockdowns that initially showed an increase in GFP intensity were followed up by observation of aggregate formation on the cuticle. RNA interference (RNAi) of collagens *dpy-3* and *col-89* showed no sfGFP::Aβ aggregates. Statistics: ordinary two-way ANOVA, error bars SEM. (**B, C**) The lack of sfGFP::Aβ aggregates could be due to a structural requirement or indirectly due to increased turnover of collagens at the extracellular matrix (ECM). To separate the two, RNAi initiated from the first larval stage (**L1**) was compared to RNAi initiated from the last (**L4**) larval stage; the latter should only take effect after the cuticle has been fully formed. (**B**) Score for aggregates as the % of the population. Statistics: ordinary two-way ANOVA, error bars SEM. (**C**) Normalized GFP intensity. Statistics: ANOVA, plotted: Tukey. (**D, E**) Knockout of *cle-1(gk364),* the ortholog of COL18A1, showed a significant reduction of sfGFP::Aβ intensity, combined

*Figure 4 continued on next page*

*Figure 4 continued*

with a mild reduction in flower aggregate formation. Statistics: ANOVA, error bars SEM. (**F**) Numbers represent individual trials for the categorized phenotype. n observed gives the total number of times the experiment was performed. RNAi of conserved collagens showed no effect on sfGFP::Aβ. (**G**) Colocalization of sfGFP::Aβ and collagen type IV/emb-9::mCherry showed the moss aggregates in the region of the hypodermis (absence of muscle tissue underneath). (**H**) Colocalization of sfGFP::Aβ and collagen type IV/emb-9::mCherry showed the flower aggregates in the region of the muscle tissue.

The online version of this article includes the following source data and figure supplement(s) for figure 4:

**Source data 1.** Individual data values, independent trials, and statistics for *Figure 4*.

**Figure supplement 1.** Pharmacological inhibition of collagen synthesis or cross-linking affects sfGFP::Aβ aggregate levels in the cuticle.

**Figure supplement 1—source data 1.** Raw data for *Figure 4—figure supplement 1*.

**Figure supplement 2.** Colocalization of sfGFP::Aβ and collagen type IV/emb-9::mCherry revealed that the moss aggregates consistently colocalize to the hypodermis in the absence of muscle tissue underneath.

**Figure supplement 3.** Colocalization of sfGFP::Aβ and collagen type IV/emb-9::mCherry revealed the flower aggregates consistently colocalize above the muscle tissue.

**Figure supplement 4.** Colocalization of sfGFP::Aβ and collagen type IV/emb-9::mCherry showed that they do not colocalize.

### ADM-2 reduces ss::sfGFP::Aβ intensity

ADM-2 is a disintegrin plus metalloprotease family member, a membrane-bound metalloprotease with extracellular peptidase M12B, disintegrin, and EGF-like domains. ADM-2 is an ortholog of the human ADAM9, which is implicated in inflammation, cancer, and AD by cleaving the APP (*Chou et al., 2020*), but whether ADAM9 plays a potential role in Aβ removal is unknown. To verify the increase of sfGFP::Aβ upon *adm-2* knockdown, we crossed a deleterious mutant for *adm-2(ok3178)* into the wild-type sfGFP::Aβ strain as well as into the *cri-2(gk314)* mutant strain. We confirmed that *cri-2* mutants showed lower, whereas *adm-2* mutants showed higher sfGFP::Aβ fluorescent levels (*Figure 6A*). The double mutants of *cri-2; adm-2* showed wild-type sfGFP::Aβ fluorescent levels (*Figure 6A*), suggesting that the benefits of losing the inhibitor of metalloprotease CRI-2 on Aβ aggregation are dependent on ADM-2.

### ADM-2 is required for Aβ aggregate removal

Next, we assessed whether the changes in GFP intensity reflect changes in secreted sfGFP::Aβ aggregates. In wild-type sfGFP::Aβ populations, about 81% showed the 'flower' structures (*Figure 6B*), and about 93% showed the 'moss' structures (*Figure 6C*). Similar to the total sfGFP::Aβ fluorescent levels at the 24 hr time point, we found that *cri-2* mutant had lower, whereas *adm-2* mutants had higher moss and flower-positive animals, which in *cri-2; adm-2* double mutants were returned to wild-type levels (*Figure 6B and C*). This confirms that the observed changes in GFP intensity reflect changes in aggregation and highlights that *adm-2* was required for the clearance of extracellular Aβ aggregates in the ECM.

### ADM-2 is sufficient for Aβ aggregate removal

To test whether ADM-2 is sufficient to reduce Aβ aggregation, we constructed a transgene with ADM-2::mScarlet-I that inducibly overexpressed ADM-2 under the control of the *hsp-16.2* promoter. The mScarlet-I fluorophore is situated at the cytoplasmic domain so as not to obstruct ADM-2 activity in the extracellular space. While the initial localization was the same as they are both induced by the same promoter, ADM-2::mScarlet-I did not colocalize with sfGFP::Aβ aggregates at 24 hr after induction (*Figure 6—figure supplement 1*). This could be either due to the mScarlet tag being intracellular, that is, only membrane-bound ADM-2 was visible, or ADM-2 efficiently removed nearby sfGFP::Aβ. ADM-2 overexpression showed a trend toward a reduced count and intensity of both moss and flower aggregates at 24 hr (*Figure 2—figure supplement 2*). However, when the aggregate count, size, and intensity were compared 48 hr after induction, we found a significant reduction of all aggregation structures upon ADM-2 overexpression (*Figure 6D–F*). These data support the idea that ADM-2 is both required and sufficient to reduce sfGFP::Aβ aggregates in the cuticle. Thus, we propose the model that TIMP(CRI-2) inhibits ADAM (ADM-2) to either directly reduce Aβ aggregation or indirectly via ECM remodeling (*Figure 6G*).

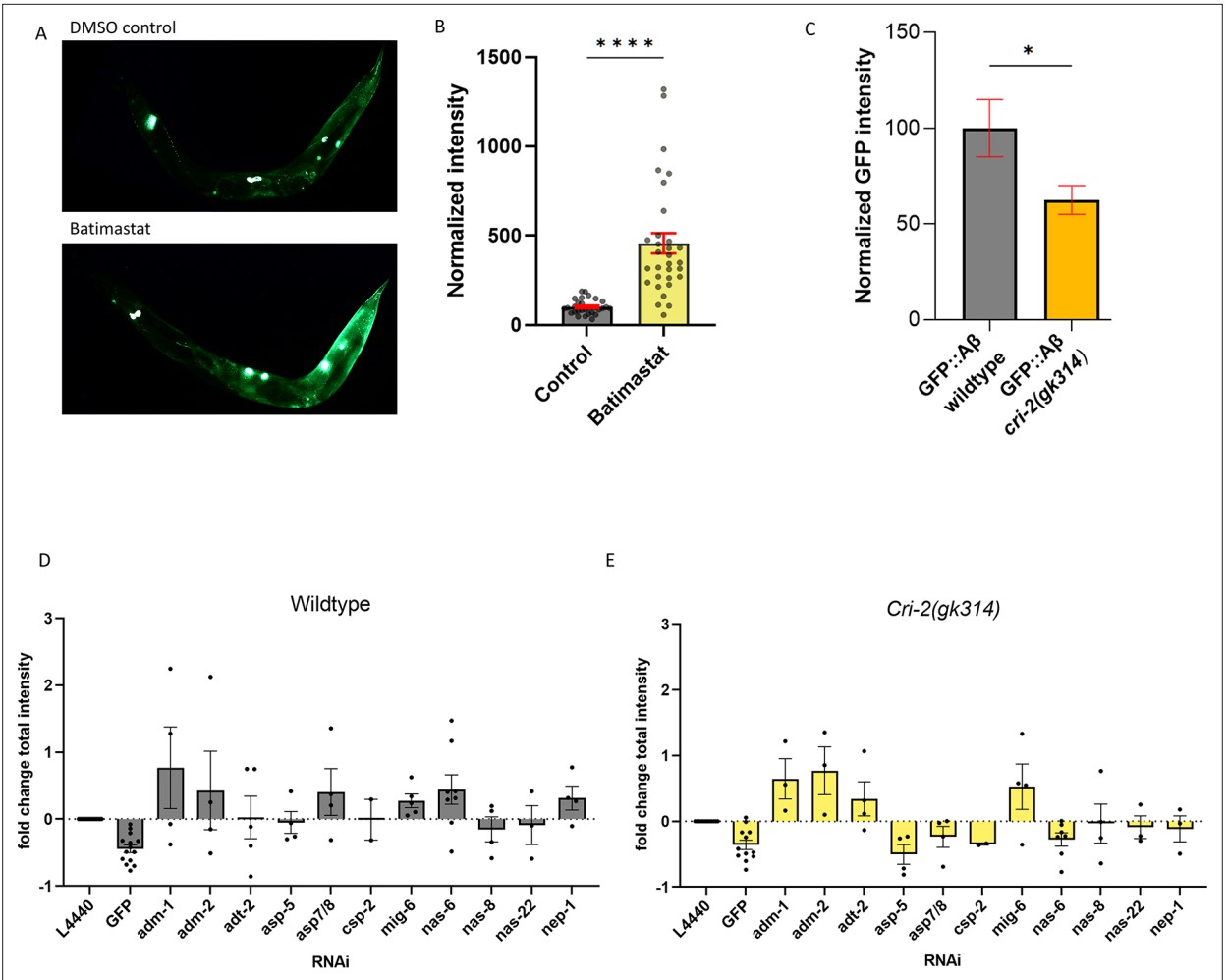

**Figure 5.** Regulation of extracellular matrix (ECM) structure and turnover influences aggregates. (**A, B**) Exposure to the metalloproteinase-inhibiting drug batimastat showed a marked increase in sfGFP::Aβ, suggesting reduced removal. (**A**) A representative image of the GFP intensity on the DMSO control treatment and a representative image of the GFP intensity on the batimastat treatment. (**B**) Combined, normalized data of three independent trials. Statistics: unpaired t-test. Error bars represent SEM. (**C**) The deletion of the tissue inhibitor of metalloproteases (TIMP) *cri-2(gk314)* showed a decrease of sfGFP::Aβ load, suggesting increased activity of metalloproteases. Statistics: unpaired t-test. Error bars represent SEM. (**D, E**) To determine which metalloprotease under the regulation of CRI-2 showed the most potential to assist in the removal of extracellular sfGFP::Aβ, GFP intensities per population were measured for RNA interference (RNAi) of several individual metalloproteases and compared between the wild-type and *cri-2* mutant background. Loss of a disintegrin and metalloprotease 2 (ADM-2) showed the largest increase in sfGFP::Aβ intensity and was selected for follow-up. Plotted: normalized mean of independent trials (each trial is one dot) with SEM.

The online version of this article includes the following source data and figure supplement(s) for figure 5:

**Source data 1.** Quantification and raw data for *Figure 5*.

**Figure supplement 1.** Secreted sfGFP::Aβ intensities were similar in induction between wild-type and *cri-2(gk314)* backgrounds, yet, at 24 hr, less GFP signal is found for the *cri-2(gk314)* background.

## Discussion

Aβ plaques are a hallmark of AD. Collagen is consistently found within these plaques, but a functional relation between Aβ and collagens has not been shown in vivo. We assessed the interaction between ECM components and regulators of ECM turnover for a potential mediating role in amyloid pathology, using novel *C. elegans* transgenic strains to express and secrete human Aβ, Aβ$_{1-42}$. This Aβ was found to form two types of extracellular aggregates associated with underlying tissue type. Targeted genetic knockdown showed mediating effects on aggregation by several extracellular proteins, including collagens and metalloproteases. A complete absence of aggregation was observed for the knockdown of *dpy-3* and *col-8* collagens. From a selection of metalloproteases, ADM-2 were found to be

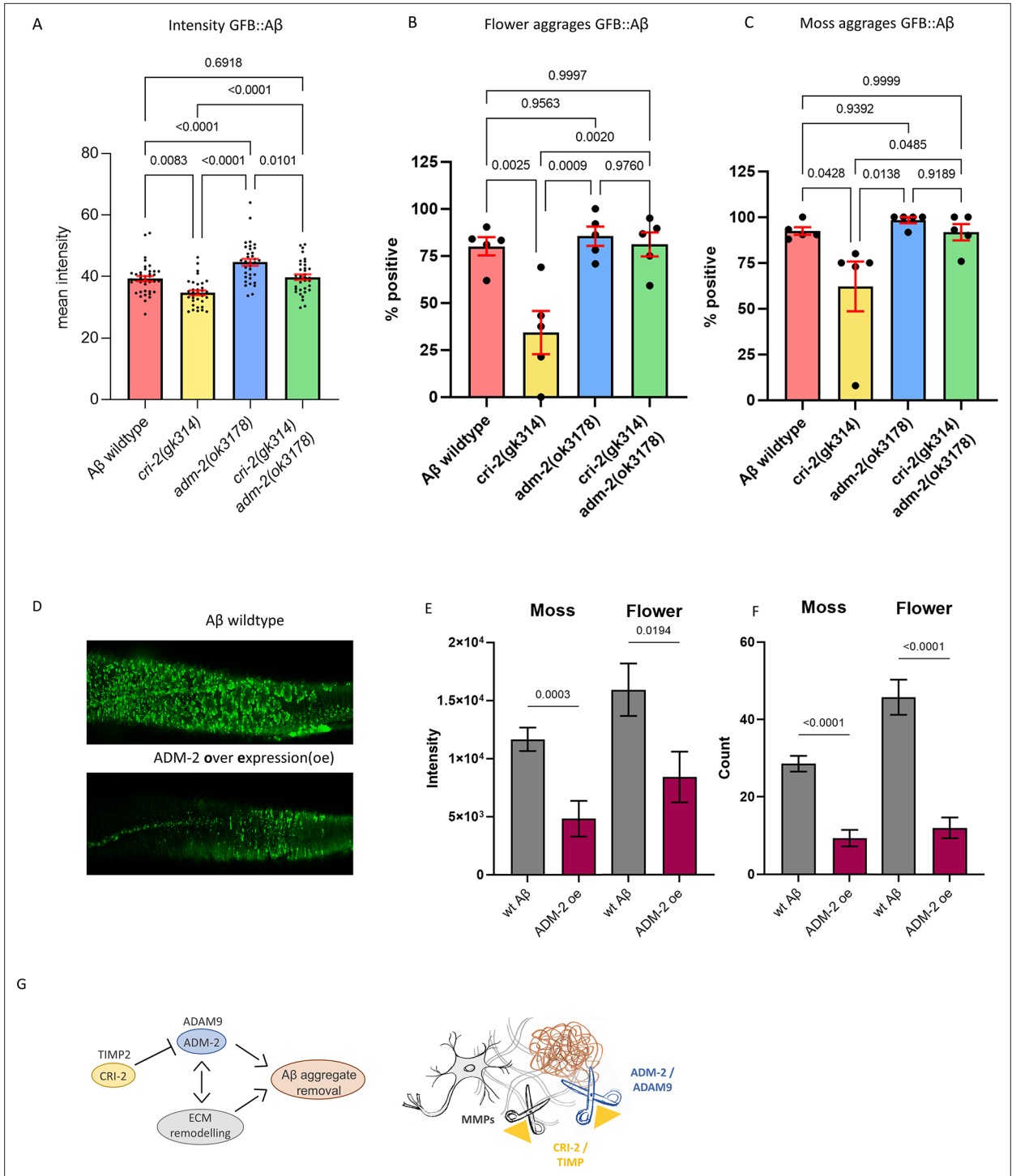

**Figure 6.** A disintegrin and metalloprotease 2 (ADM-2) was required and sufficient to remove extracellular sfGFP::Aβ. (**A**) The GFP intensity for sfGFP::Aβ was reduced in the *cri-2* mutant background. Loss of *adm-2* increased sfGFP::Aβ intensity, even in the *cri-2/adm-2* double mutant. Data were combined from four independent experiments. Statistics: ordinary two-way ANOVA. Plotted bars are mean with SEM. (**B, C**) Similarly, loss of *cri-2* resulted in fewer animals with flower and moss aggregations, while loss of *adm-2* increased both aggregation types. Loss of *cri-2* and *adm-2* in the double mutant background showed that other metalloproteases could not compensate for the loss of *adm-2* regarding the removal of extracellular aggregates, suggesting that *adm-2* was required for the removal of extracellular sfGFP::Aβ aggregates. Statistics: ANOVA. Plotted is the percentage of the population positive for aggregation type, one dot per experiment, bars are mean with SEM. (**D**) Visual representation of the observation that overexpression of *adm-2* (ADM-2 oe) led to a reduction of extracellular sfGFP::Aβ aggregates. (**E, F**) Overexpression of ADM-2 was sufficient to lead to a significant reduction of sfGFP::Aβ aggregates, both in count and intensity of aggregates, measured 48 hr after induction of transgene expression. Data

*Figure 6 continued on next page*

*Figure 6 continued*

is the intensity and count measures over two independent experiments. Plotted are mean and SEM. Statistical analysis: unpaired t-test. (**G**) Schematic representation of the mechanisms in which ADM-2, under regulation by CRI-2, either directly or indirectly assisted in removing extracellular sfGFP::Aβ aggregates.

The online version of this article includes the following source data and figure supplement(s) for figure 6:

**Source data 1.** Quantification and raw data for *Figure 6*.

**Figure supplement 1.** Intracellular ADM-2::mScarlet-I did not colocalize with sfGFP::Aβ.

**Figure supplement 2.** Overexpression of a disintegrin and metalloprotease 2 (ADM-2) is sufficient to lead to a significant reduction of sfGFP::Aβ aggregates.

most effective in reducing Aβ aggregation. We found overexpression of ADM-2 to be sufficient to remove extracellular Aβ aggregates in vivo. These findings support a potential active, mediating role for ECM components on Aβ aggregation.

Generally, metalloproteases are known for their function in mediating ECM remodeling by cleaving collagens. While some collagens are long-lived ECM components, ECM turnover allows for damage repair (wound healing) or in response to other tissue demands, such as exercise (*Ewald, 2020*; *Kritikaki et al., 2021*; *Xue and Jackson, 2015*). In combination with the collagens we found, this could imply that ADM-2 assists in the removal of Aβ aggregates by remodeling the ECM. The collagen components that, when the expression is knocked down, suppress aggregate formation suggest that there are ECM composition requirements for aggregates to form. It is unclear if these specific collagens are required for direct interaction with Aβ before aggregate formation or if a more general structural composition within the ECM is required. However, these data support the concept that ECM dynamics are fundamental to Aβ aggregation, and ECM remodeling contributes to aggregate removal.

Another category identified in the RNAi screen bridges the extracellular and intracellular environments and involves clathrin-mediated endocytosis, the sorting endosome, and (targeted) vesicle secretion. In our screen, the knockdown of RAB-7 led to an abundant accumulation of small, bright green vesicles near the cuticle, while aggregates remained absent (*Figure 3—figure supplement 1*). Furthermore, members of the ESCRT complex, as well as SNAREs and sorting nexins, were indicated to alter the localization, accumulation, and aggregate formation of Aβ. Conceptually, ECM remodeling and vesicle uptake and secretion may serve a common purpose regarding dynamic ECM adaptations. In the process of ECM turnover, metalloproteases are actively cleaving ECM components such as collagens and fibronectins (*Shi and Sottile, 2011*), and the resulting cleaved products are internalized via receptor-mediated phagocytosis and degraded in the lysosome (*Arora et al., 2000*). Furthermore, to secrete metalloproteases, newly synthesized collagens, or Aβ to the ECM, the sorting endosome and (vesicle) secretion pathways are in play (*Chang et al., 2021*). As such, these seemingly distinct mechanisms could all work together for the collective purpose of extracellular aggregate removal.

In recent work, ADM-2 overexpression was shown to lead to molting defects (*Joseph et al., 2021*). To allow growth, the cuticle of *C. elegans* is shed and replaced by a new cuticle, secreted and deposited by hypodermal cells underneath the cuticle. Initiation of the molt requires the internalization of sterol hormones and activating a cascade of proteases to mediate the shedding of the old cuticle. As such, molting depends on ECM remodeling, in which ADM-2 plays an essential role. Interestingly, one of the potential targets of ADM-2 cleavage revealed in that work is LRP-1, the *C. elegans* low-density lipoprotein receptor orthologous to human LRP1 (*Joseph et al., 2021*). LRP-1 in *C. elegans* is a membrane-bound receptor, which can sequester sterols from the extracellular environment, and when internalized together, these sterols can initiate molting (*Yochem et al., 1999*).

ADM-2 is suggested to cleave LRP-1 and release it from the membrane, then referred to as sLRP. Although this sLRP-1 can still capture sterols, they are not internalized, leading to incomplete shedding of the cuticle (*Joseph et al., 2021*). Genes involved in *C. elegans* molting that are a hit in our screen are *dab-1*, *hgrs-1*, and *apl-1*. DAB-1 is a cytoplasmic adaptor protein involved in endocytosis. Endocytosis of sterols is essential for molting (*Lažetić and Fay, 2017*). HGRS-1 is a Vps27 ortholog, which recruits ESCRT machinery to endosomes. Inhibition of HGRS-1 leads to molting defects (*Lažetić and Fay, 2017*). Interestingly, HGRS-1 and ADM-2 colocalize (*Joseph et al., 2021*), which suggests they may be involved in similar pathways through direct interaction. Loss of APL-1, the APP ortholog,

causes lethal defects upon shedding the cuticle, which is rescued by the expression of the extracellular part of APL-1 (*Hornsten et al., 2007*). The association of multiple genes involved in the molting process with an increase in sfGFP::Aβ load and aggregate formation in adult *C. elegans* suggests that changes in ECM dynamics can influence amyloid aggregate load. However, the exact role of ADM-2 and its potential targets needs further refining.

The human ortholog of ADM-2, ADAM9, has been implicated in AD and is suggested to regulate the shedding of APP as an alpha-secretase, either indirectly by regulating ADAM10 or by functioning as an alpha-secretase itself, cleaving APP in a non-amyloidogenic manner (*Asai et al., 2003*; *Moss et al., 2011*). The cleavage site for alpha-secretase is situated in the middle of the Aβ fragment, potentially allowing direct cleavage of Aβ peptides. Moreover, human ADAM9 can be alternatively spliced, losing its transmembrane and cellular domains, resulting in an extracellular, active enzyme (*Hotoda et al., 2002*). This could potentially be a way for ADM-2 to reach the Aβ aggregates in the cuticle. As an ortholog of ADAM9, ADM-2 could potentially cleave Aβ directly and, as such, assist in the removal of extracellular aggregates.

In conclusion, we established an in vivo model to trace Aβ aggregation in the ECM. Our findings suggest that activating ECM remodeling promotes aggregate removal and could become an important strategy to ameliorate AD disease progression.

## Materials and methods
### Strain handling
Preparation of NGM agar plates, feeding with OP50 *Escherichia coli,* and handling of *C. elegans* strains by picking as described by *Stiernagle, 2006*. For maintenance of *C. elegans* strains, 10 adults are picked and transferred to a fresh plate per generation and are kept at 15°C.

### *C. elegans* strains
For the generation of the strain expressing the transgene sfGFP::Aβ, the germline of *C. elegans* N2 Bristol (wild-type) was injected with the plasmid pLSD134 at 50 ng/µL and pRF4 *rol-6(su1006)*, also at 50 ng/µL. The total concentration of DNA in the injection mix was 100 ng/µL. Plasmid pLSD134 was cloned by VectorBuilder. Detailed plasmid sequence and map are in *Supplementary file 4*. The extrachromosomal array was integrated into the genome using UV irradiation with the Stratagene UV Stratalinker 2400 (254 nm). The resulting integrated strain was backcrossed with N2 Bristol four times and named LSD2104, which was used throughout this study. LSD2104 *xchIs015* [pLSD134 P*hsp-16.2*::ssSel1::FLAG::superfolderGFP::spacer::humanAmyloidBeta1-42::let-858–3'UTR; pRF4 *rol-6(su1006)*].

To attain deletion of the genes *cri-2, adm-2, col-99 and cle-1* in LSD2104(sfGFP::Aβ), LSD2104 was crossed to VC718 *cri-2(gk314) V*, RB2342 *adm-2(ok3178) X*, RB1165 *col-99(ok1204) IV,* and VC855 *cle-1(gk364) I*; resulting in the strains LSD2165, LSD2201, LSD1056, and LSD1052, respectively. The double mutant background with deletions for *cri-2(gk314)* and *adm-2(ok3178)* was generated by crossing LSD2165(sfGFP::Aβ, *cri-2(gk314) V*) with RB2342 *adm-2(ok3178) X*, resulting in the strain LSD2204.

To induce overexpression of ADM-2 in LSD2104(sfGFP::Aβ) background, LSD2104 was injected with the plasmid pLSD170, *hsp-16.2p*::adm-2::mScarlet-I. This plasmid was designed to express ADM-2 with the mScarlet-I tag in the cellular compartment so as not to obstruct enzymatic function. Plasmid pLSD170 was cloned by VectorBuilder, and a detailed plasmid sequence and map are in *Supplementary file 4*. Injection of pLSD170 was performed with a total DNA concentration of 50 ng/µL. Exposure to UV to induce integration was not successful, and a non-integrated line, not exposed to UV, was maintained by selecting for expression of ADM-2::mScarlet-I, resulting in the strain LSD3014.

More details on strains and primers for the identification of strains are in *Supplementary file 4*.

### Induction of ss::sfGFP::Aβ expression
Age-synchronized populations, as described by Teuscher (*Teuscher et al., 2019b*), were grown at 20°C on NGM plates for 4 days until the young-adult stage. The heat shock was performed by placing the plates at 33°C for 2 hr, after which they were returned to 20°C. The assessment was 24 hr after heat-shock induction unless otherwise mentioned.

## Assessment of GFP intensity

The intensity of the GFP signal was obtained from imaging with an upright bright-field fluorescence microscope, camera, and filter set, as described in *Teuscher et al., 2019a*. Analysis software used is Fiji (*Schindelin et al., 2012*), with a program described in GitHub (*Statzer et al., 2021*), code accessible on https://github.com/JongsmaE/GreenIntensityCalculator ( *Jongsma, 2021*). Briefly, the triple filter set is used to separate autofluorescence in the *C. elegans* gut from the GFP signal, and the autofluorescence appears yellow. The GFP intensity is calculated by the program as follows: the color image is split into green, blue, and red channels. Since yellow is an addition of green and red, green pixels are only counted if 'green intensity' > 'red intensity', and the red value is subtracted from the green value. The remaining green values are added up per selected area (worm) to obtain the total intensity. The number of pixels is counted as well to calculate the average intensity per pixel, in case one wants to account for animal size. A minimum of 20 animals are measured per condition.

## Western blot

Aggregates were heat-induced 24 hr post-L4 stage in sfGFP::Aβ transgenic animals, and 24 hr post-induction, animals were collected in the M9 buffer and washed three times with the M9 buffer to remove excess bacteria. The pellet was flash-frozen and stored at –80°C. Protein was isolated from the pellet by homogenizing it in Buffer 'C' solution (composition: HEPES 50 mM, KCl 100 mM, MgCl$_2$ 1 mM, EGTA 1 mM, glycerol 10%) containing protease inhibitor cocktail (Merck #11873580001). Protein estimation was done using Bradford assay (Bio-Rad #5000001). Approximately 30 μg of protein was run in triplet SDS-PAGE gels. After blotting the protein on the PVDF membrane (Millipore #IPVH00010), the membranes were blocked in a 5% BSA blocking buffer for 1 hr at room temperature while shaking. Primary antibodies - anti-Aβ peptide (MOAB-2) pan antibody, clone 6C3 (Merck #MABN254), and anti-GFP (Roche #11814460001) - were used in dilution 1:5000, and monoclonal anti-α-tubulin antibody (Sigma #T9026) was used in dilution 1:10,000 in the blocking buffer overnight at 4°C. Next day, the blots were washed three times with 1X TBST buffer and put in secondary anti-mouse IgG HRP-linked antibody (#7076) in the dilution 1:10,000 for 1 hr at room temperature. Then, the blots were washed three times with TBST buffer and developed in a chemiluminescent chamber using Clarity Western ECL substrate (#1705061) was used.

## RNAi screen

RNAi plates are prepared as described before with the addition of carbenicillin (50 μg/mL) and isopropyl-D-1-thiogalactopyranoside (IPTG) (1 mM) after autoclaving. These plates are seeded with bacteria carrying RNAi. These bacteria originate from the Vidal RNAi library (*Rual et al., 2004*) and Ahringer RNAi library (*Fraser et al., 2000*; *Kamath et al., 2003*). Clones were copied from the Vidal and Ahringer libraries by growth overnight at 37°C on an LB-agar plate containing carbenicillin (50 μg/mL) and tetracycline (12.5 μg/mL) (carb/tet) and consequently grown in liquid LB(+carb/tet) from a single colony. Sequence-confirmed glycerol stocks were stored at –80°C. For RNAi experiments, clones are selected from the frozen library at –80°C and grown overnight in liquid LB with ampicillin 50 μg/mL and tetracycline (12.5 μg/mL). The following day, the cultures were spun down, and the LB(amp/tet) medium was refreshed and filled to 5× the original volume. The cultures are allowed to grow for 3 hr, aiming to harvest them in the growth phase. They are then concentrated 20× and resuspended in LB supplemented with 1 mM IPTG to induce replication of the RNAi. These were then seeded onto the 6 cm RNAi plates, 500 μL each. Approximately 40 animals were allowed to feed at 20°C from larval stage 1 to young adulthood before heat-shock induction. As a negative control, the empty vector pL4440 was used. As a control for RNAi induction, a vector carrying RNAi for GFP is used. For the RNAi of genes that led to developmental delay, RNAi was repeated from the L4 stage to re-assess an effect on sfGFP::Aβ intensity. Raw data are presented in *Supplementary file 1*, together with the list of hits, including developmental delay, in separate tabs of the file.

## Pharmacological treatments

Broad spectrum matrix metalloprotease (MMP) inhibitor - Batimastat. A stock concentration of 50 mM Batimastat (also known as BB-94; Sigma-Aldrich #SML0041-5MG) was made by dissolving 5 mg in 0.210 mL DMSO 100%. 300 μL of a 50× dilution in M9 was added to the (diameter 10 cm) NGM plates with young adults (at a concentration of 1 mM BB-94 and 2% DMSO) 1 hr prior to the

heat-shock induction. The control group was treated with the same volume of M9, 2% DMSO. The strains then remained in this environment until imaging 24 hr after the heat-shock induction of sfGF-P::Aβ expression.

Collagen inhibitors: A stock concentration of 500 mM was made for both drugs. 780 mg BPY (Sigma #1030980005) was dissolved in 10 mL 100% DMSO, and 640 mg 3-aminopropionitrile fumarate (BAPN, Sigma, #A3134) was dissolved in 10 mL $H_2O$. These solutions were diluted further in $H_2O$ 20 times to make 25 mM. 100 µL of this 20× diluted solution was top-coated on the L4 staged animals in a 25 mL NGM containing Petri plates to make the final working concentration 100 µM. 24 hr later, the animals were heat-shocked at 33°C for 3 hr and kept back at 15°C for aggregate induction. Imaging was done 24 hr post-heat-shock induction.

## Confocal imaging and FRAP

Images of aggregation were taken using an upright confocal laser scanning microscope as described by *Hess et al., 2019*. Adaptations: 60/1.00 oil objective, excitation at 488 nm, an intensity of 0.3%, and a 2% agarose pad. Additionally, photobleaching was performed by exposing a selected region to a UV (405 nm) laser at 3% intensity for 17 s. Photobleached areas were observed for recovery of fluorescent signal from the first seconds up to 4 hr after photobleaching.

## Correlative light and electron microscopy

*C. elegans* were fixed with 4% formaldehyde and 0.1% glutaraldehyde in 0.1 M sodium cacodylate buffer, immersed in gelatine 12%, cryoprotected with 2.3 M sucrose. Samples were frozen in liquid nitrogen, and using an ARTOS 3D ultracut system equipped with a cryochamber EM UC7 Leica, 110 nm ultrathin cryosections were collected on 7×7 $mm^2$ silicon wafers with fluorescent beads (PS-Speck, ThermoFisher). Light microscopy images were acquired with a wide-field microscope, Thunder Leica, objective 100×/1.44. Electron microscopy images from the very same section were taken with a SEM Auriga 40 Zeiss microscope at an acceleration voltage of 800 eV, with an InLens detector, pixel size 4 nm, and dwell time 100 µs. Registration and alignment of the light and electron microscopy images were done with TrakEM2 (*Cardona et al., 2012*) within the open-source platform Fiji (*Schindelin et al., 2012*).

## Acknowledgements

We thank Victoria Brügger for her help with screening and Charlotte Meneghin for helping with the validation of the screen hits. Some *C. elegans* strains were provided by the CGC, which is funded by the NIH Office of Research Infrastructure Programs (P40 OD010440). We thank David S Fay for the helpful discussions. We thank Ursula Lüthi for cryosectioning. EJ was funded for the first 3 years by a grant from the Mibelle Group Biochemistry, Mibelle AG, Switzerland. Funding from the Swiss National Science Foundation PP00P3_163898 and 190072 to CYE and 190072 to EJ and AG.

## Additional information

### Funding

| Funder | Grant reference number | Author |
| --- | --- | --- |
| Schweizerischer Nationalfonds zur Förderung der Wissenschaftlichen Forschung | PP00P3_163898 | Collin Yvès Ewald |
| Schweizerischer Nationalfonds zur Förderung der Wissenschaftlichen Forschung | 190072 | Elisabeth Jongsma Anita Goyala Collin Yvès Ewald |

| Funder | Grant reference number | Author |
|---|---|---|

The funders had no role in study design, data collection and interpretation, or the decision to submit the work for publication.

## Author contributions

Elisabeth Jongsma, Conceptualization, Resources, Data curation, Formal analysis, Supervision, Funding acquisition, Validation, Investigation, Visualization, Methodology, Writing – original draft, Project administration; Anita Goyala, Formal analysis, Investigation, Visualization, Writing – review and editing; José Maria Mateos, Data curation, Formal analysis, Validation, Methodology, Writing – review and editing; Collin Yvès Ewald, Conceptualization, Resources, Data curation, Supervision, Funding acquisition, Visualization, Methodology, Writing – original draft, Project administration, Writing – review and editing

## Author ORCIDs
Anita Goyala ⓘ http://orcid.org/0000-0001-9196-2915
José Maria Mateos ⓘ https://orcid.org/0000-0002-3675-6198
Collin Yvès Ewald ⓘ http://orcid.org/0000-0003-1166-4171

## Decision letter and Author response
Decision letter https://doi.org/10.7554/eLife.83465.sa1
Author response https://doi.org/10.7554/eLife.83465.sa2

---

# Additional files

## Supplementary files
• Supplementary file 1. RNA interference (RNAi) assessed genes and libraries. Individual measurements, independent biological trials, raw and processed data, and statistics are provided.

• Supplementary file 2. RNA interference (RNAi) hits and categories. Individual measurements, independent biological trials, raw and processed data, and statistics are provided.

• Supplementary file 3. Ortholog collagens data overview. Detailed overview about the collagens orthologs.

• Supplementary file 4. Reagents list. Detailed lists of *C. elegans* strains, bacterial strains, RNA interference (RNAi) clones, plasmids, and primers are provided.

• MDAR checklist

## Data availability
All source data, raw data, screening hits, plasmid sequences, and statistical information are provided in Data Source Files and in Supplementary Files 1-4.

---

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
