## [Editor Report]

It is of significance to generate a novel transgenic *C. elegans* model with inducible expression and secretion of human GFP-tagged human Aβ1-42 for the neurodegenerative disease field. This paper is interesting to neuroscientists who work on protein aggregation and neurodegenerative diseases. The key claims of the manuscript are well supported by the revised convincing data revealing that the metalloprotease ADM-2 modulates ECM and assists in the removal of extracellular Aβ aggregates.

---

## [Decision Letter]

**Decision letter after peer review:**

Thank you for submitting your article "Removal of extracellular human amyloid β aggregates by extracellular proteases in *C. elegans*" for consideration by *eLife*. Your article has been reviewed by 3 peer reviewers, one of whom is a member of our Board of Reviewing Editors, and the evaluation has been overseen by Jeannie Chin as the Senior Editor. The following individual involved in review of your submission has agreed to reveal their identity: Evandro Fei Fang (Reviewer #3).

Essential revisions:

Although the reviewers found that the current study is of interest, they all raise major concerns that need to be addressed. Some of the main concerns are listed as below:

1) It remains unclear how ADM-2 removes extracellular Abeta aggregates.

2) It is unknown whether collagens escalation increases aggregate formation.

3) There is no direct evidence to prove that the GFP-positive aggregates do contain human Aβ. This can be achieved by methods including: Western blot to show aggregated Aβ, electron microscopy to observe Aβ fibrils, and immunostaining showing the colocalization between GFP fluorescence and human Aβ.

4) More detailed mechanistic studies, as well as further characterization of the strain, is necessary (e.g., whether this strain shows impaired memory, the key feature of AD-using the well developed chemotaxis assay)

*Reviewer #1 (Recommendations for the authors):*

The concerns for the current manuscript are: (1) it remains unclear how ADM-2 removes extracellular aggregates. (2) it is missing whether collagens escalation increases aggregate formation.

These two biological processes are critical for understanding the balance in Abeta aggregate formation.

*Reviewer #2 (Recommendations for the authors):*

In this manuscript, the authors used new models to study the role of ECM on Aβ aggregation and clearance. I have some suggestions to make the claims stronger.

1. The quantification of Aβ aggregation is not solid enough, GFP fluorescence intensity and morphology is not sufficient to show the presence of Aβ aggregates.

2. There is no direct evidence to prove that the GFP-positive aggregates do contain human Aβ. This can be achieved by methods including Western blot to show aggregated Aβ, electron microscopy to observe Aβ fibrils, and immunostaining showing the colocalization between GFP fluorescence and human Aβ.

3. There should be evidence that the RNAi did work to knock down the target genes by PCR or Western blots.

4. The overexpression of ADM-2 should be confirmed.

*Reviewer #3 (Recommendations for the authors):*

The authors generated a novel sfGFP::Aβ *C. elegans* models of AD that expresses Abeta aggregates extracellularly; using this worm model, they identified that a disintegrin and metalloprotease ADM-2, an ortholog of human ADAM9, participated in removing these extracellular aggregates. This worm model may be very useful to the AD field. Addressing the below questions will improve the quality of the paper.

– While the major focus of this paper was on the elimination of Abeta plaques, it is important to give a summary on the multiple causes/risks of AD as elimination of Abeta plaques may not be sufficient to inhibit AD (the recent Abeta antibody by the Swedish company showing 28% reduction of memory loss via an Abeta antibody while significant side effects reported). Pathologies features like Tau tangles, defective mitophagy, senescence, and inflammation should be mentioned (PMID: 23254930; PMID: 30742114; PMID: 30936558). The authors may not necessarily have to test whether ADM-2 shows any direct/indirect/non-autonomous effects on Tau pathologies, defective mitophagy, senescence, and inflammation, it could be nice for them to discuss such things in the 'Discussion' section.

– Any evidence to validate that the sfGFP::Aβ strain expresses Abeta1-42, rather than Abeta3-42 (as happened in other Abeta strains before)?

– Did the single heat shock induction (33 C for 2 h) induce signalling pathways that may contribute to 'artificial effects' in Abeta clearance?

– As AD pathologies are primarily in the brain region, a good worm model should also mimic this feature to enable a higher likelihood of data interpretation from worms to humans. As the 'nerve ring' is like the 'brain', have the authors considered expressing Abeta plaques (primarily) in this region?

– The authors noticed an important phenomenon that 'different collagens to ameliorate or potentiate Aβ aggregation': what are the underlying molecular mechanisms? It could be nice to discuss underlying mechanisms in the 'Discussion section'.

---

## [Author Response]

Essential revisions:Although the reviewers found that the current study is of interest, they all raise major concerns that need to be addressed. Some of the main concerns are listed as below:1) It remains unclear how ADM-2 removes extracellular Abeta aggregates.

It turned out to be difficult to nail the exact mechanism of how ADM-2 removes extracellular Abeta aggregates.

In our discussion, we propose two possible mode-of-action (MoA) for ADM-2, direct and indirect potential mechanisms, which are not mutually exclusive.

Potential direct MoA for ADM-2:

The human ortholog of ADM-2, ADAM9, has been suggested to act as an α-secretase itself, cleaving APP in a non-amyloidogenic manner (Asai et al., 2003; Moss et al., 2011). In Author response image 1, we show the only evidence for this hypothesis we had obtained by ECM-enriching Western blotting, a protocol adapted from Della David’s Lab (doi: 10.3791/56464). In brief, we prepared *C. elegans* samples into soluble and insoluble fractions and used higher Urea concentrations to solubilize the ECM. The predicted cleavage based on human ADAM9 and APP (https://doi.org/10.1016/S0006-291X(02)02999-6, doi: 10.1074/jbc.M111.280495) is after amino acid 17 in Abeta(1-42), resulting in the cleaving-off of a 2.5 kDa fragment of our sfGFP::Aβ (Author response image 1). By western blot, we detected these two predicted sizes of full-length sfGFP::Aβ (44.6 kDa) and cleaved sfGFP::Aβ (42.2 kDa) (Author response image 1). In both wild type (N2) and sfGFP::Aβ(F20S, L35P) non-aggregation forming (AβΔ), we do not detect any GFP tagged Abeta, suggesting that we look at the aggregated and insoluble Abeta (potentially from the ECM) (Author response image 1). The sfGFP::Aβ (Aβ) showed a similar ratio of full-length sfGFP::Aβ (44.6 kDa) and cleaved sfGFP::Aβ (42.2 kDa) (Author response image 1, C). In cri-2 mutants (cri-2(gk314); sfGFP::Aβ (Aβ cri-2)), the cleaved Abeta is higher, consistent with the release of inhibition and higher activity of metalloproteases (Author response image 1, C). The adm-2 mutants (adm-2(ok1204); sfGFP::Aβ (Aβ adm-2), seem to have higher Abeta levels with a neglectable reduction in cleaved Abeta (Author response image 1, C). However, the double mutation cri-2; adm-2 (adm-2(ok1204); cri-2(gk314); sfGFP::Aβ (Aβ double) has higher Abeta levels, but the ratio of cleaved vs full-length is not higher as seen in single mutant cri-2 (Author response image 1, C), consistent with the model that CRI-2 inhibits ADM-2 function for cleaving Abeta and removal of aggregates levels.

Although we find this observation intriguing, to really prove this, further experiments are required, such as Mass-spec verification of fragments and generating an “adm-2-non-cleavable human Abeta”. For this, protocol development and other resources are required that are beyond the scope of this study but should be addressed in a new project. Therefore, we leave this tantalizing preliminary data here in the response letter and only propose this possibility in the discussion of a potential direct MoA for ADM-2.

Discussion

“The human ortholog of ADM-2, ADAM9, has been implicated in AD and is suggested to regulate the shedding of APP as an α-secretase, either indirectly by regulating ADAM10 or by functioning as an α-secretase itself, cleaving APP in a non-amyloidogenic manner (Asai et al., 2003; Moss et al., 2011). The cleavage site for α-secretase is situated in the middle of the Aβ fragment, potentially allowing direct cleavage of Aβ peptides. Moreover, human ADAM9 can be alternatively spliced, losing its transmembrane and cellular domains, resulting in an extracellular, active enzyme (Hotoda et al., 2002). This could potentially be a way for ADM-2 to reach the Aβ aggregates in the cuticle. As an ortholog of ADAM9, ADM-2 could potentially cleave Aβ directly and, as such, assist in the removal of extracellular aggregates.”

**Author response image 1. sa2fig1:** Potential direct MoA for ADM-2 in Abeta cleavage. (A) Amino acid sequence of human Abeta with predicted cleaved-off fragment (in red) after ADAM9 cleavage. For our transgenic fusion protein of human Abeta (1-42) with super-folder GFP (sfGFP::Aβ) would result in predicted sizes of full-length sfGFP::Aβ: (B) 44.6 kDa, cleaved sfGFP::Aβ: 42.2 kDa, pure GFP: 27 kDa. (C) Western blot. Strains: wild type (N2), LSD1091 sfGFP::Aβ(F20S, L35P) non-aggregation forming (AβΔ), LSD2104 sfGFP::Aβ (Aβ), LSD2165 cri-2(gk314); sfGFP::Aβ (Aβ cri-2), LSD2201 adm-2(ok1204); sfGFP::Aβ (Aβ adm-2), LSD2204 adm-2(ok1204); cri-2(gk314); sfGFP::Aβ (Aβ double). Antibody: 1st anti-GFP (mouse)2nd anti-mouse-HRP. Sample preparation adapted from Groh et al., 2017 doi: 10.3791/56464: 1. Collect worm pellet and lyse with tissue lyser according to lab protocol. 2. Centrifuge at 18,400 x g for 20 min at 4 °C. Collect the supernatant. 3. Resuspend the pellet in 100 µL RIPA buffer and centrifuge at 18,400 x g for 20 min at 4°C. Discard the supernatant, spin shortly, and be careful to remove leftover supernatant. 4. To solubilize the highly insoluble proteins, resuspend the pellet in 75 µL Urea/SDS buffer (8 M urea, 2% SDS, 50 mM dithiothreitol (DTT), 50 mM Tris, pH 8) and incubate for 10 min at room temperature. (C) Relative density. The LSD2104 sfGFP::Aβ (Aβ) was taken as the normal reference point to build the ratio of full-length sfGFP::Aβ (44.6 kDa) vs cleaved sfGFP::Aβ (42.2 kDa).

Potential indirect MoA for ADM-2:

A second way of ADM-2 action might be through either regulation molting or directly cleaving collagens, both resulting in ECM remodeling and, thereby, excision of the Abeta aggregates from the cuticle. Based on previous findings from Joseph et al., 2021, we discuss this in two paragraphs in the discussion.

“In recent work, ADM-2 overexpression was shown to lead to molting defects (Joseph et al., 2021). To allow growth, the cuticle of *C. elegans* is shed and replaced by a new cuticle, secreted and deposited by hypodermal cells underneath the cuticle. Initiation of the molt requires the internalization of sterol hormones and activating a cascade of proteases to mediate the shedding of the old cuticle. As such, molting depends on ECM remodeling, in which ADM-2 plays an essential role. Interestingly, one of the potential targets of ADM-2 cleavage revealed in that work is LRP-1, the *C. elegans* low-density lipoprotein receptor orthologous to human LRP1 (Joseph et al., 2021). LRP-1 in *C. elegans* is a membrane-bound receptor, which can sequester sterols from the extracellular environment, and when internalized together, these sterols can initiate molting (Yochem et al., 1999).

ADM-2 is suggested to cleave LRP-1 and release it from the membrane, then referred to as sLRP. Although this sLRP-1 can still capture sterols, they are not internalized, leading to incomplete shedding of the cuticle (Joseph et al., 2021). Genes involved in *C. elegans* molting that are a hit in our screen are dab-1, hgrs^-1^, and apl-1. DAB-1 is a cytoplasmic adaptor protein involved in endocytosis. Endocytosis of sterols is essential for molting (Lažetić and Fay, 2017). HGRS^-1^ is a Vps27 ortholog, which recruits ESCRT machinery to endosomes. Inhibition of HGRS^-1^ leads to molting defects (Lažetić and Fay, 2017). Interestingly, HGRS^-1^ and ADM-2 colocalize (Joseph et al., 2021), which suggests they may be involved in similar pathways through direct interaction. Loss of APL-1, the APP ortholog, causes lethal defects upon shedding the cuticle, which is rescued by the expression of the extracellular part of APL-1 (Hornsten et al., 2007). The association of multiple genes involved in the molting process with an increase in sfGFP::Aβ load and aggregate formation in adult *C. elegans* suggests that changes in ECM dynamics can influence amyloid aggregate load. However, the exact role of ADM-2 and its potential targets needs further refining.”

To test this idea of indirect effects on Abeta aggregates levels by ECM changes, we rationalized to use of a red fluorescent tagged collagen marker (Author response image 2). We crossed the LSD2104 sfGFP::Aβ strain with a CRISPR-inserted wormScartlet into cuticular collagen rol-6 (ROL-6::mScarlet, a generous gift from Cathy Savage-Dunn; https://doi.org/10.7554/*eLife*.75906). The cuticular collagen ROL-6::mScarlet appears beneath the sfGFP::Aβ (Author response image 3), consistent with our confocal images where sfGFP::Aβ is sandwiched between the two cuticle layers. Next, we knock down several collagen hits from our screen that either increased or decreased sfGFP::Aβ levels, including col-119, col-136, and col-137 (shown in Author response image 4), dpy-7 (shown in Author response image 5), col-146, col-147, col-153, col-164, and col-180 (not shown). We found that the knockdown of col-119, col-136, and col-137 cuticular collagens affects sfGFP::Aβ aggregates but not the ECM morphology represented by collagen ROL-6::mScarlet. By contrast, dpy-7 RNAi altered ECM morphology represented by collagen ROL-6::mScarlet and resulted in no extracellular sfGFP::Aβ aggregates (Author response image 5). These results indicate dependent and independent mechanisms of ECM changes on extracellular sfGFP::Aβ aggregates.

**Author response image 2. sa2fig2:** Collagen ROL-6::mScarlet. Representative image of ROL-6::mScarlet pattern both underneath the muscle and hypodermal tissues. A) GFP channel B) mScarlet-I channel C) DCI D) combined. Scale bars 10μm.

**Author response image 3. sa2fig3:** No co-localization of sfGFP::Aβ and collagen ROL-6::mScarlet. No co-localization is observed. The cuticle collagen ROL-6 tagged with wrmScarlet-I (codon-optimized mScarlet fluorophore) showed no obvious difference in pattern nor brightness between wildtype and Aβ expressing strains. Scale bars 10μm.

**Author response image 4. sa2fig4:** Knockdown of different cuticular collagens affects sfGFP::Aβ aggregates but not the ECM morphology represented by collagen ROL-6::mScarlet. (**A**,**B**) Knockdown of col-119 by RNAi resulted in a slightly lower number of sfGFP::Aβ but no obvious effect on the collagen structure of ROL-6::mScarlet. (**C**,**D**) Knockdown of col-136 by RNAi resulted in a lower number of sfGFP::Aβ but no obvious effect on the collagen structure of ROL-6::mScarlet. (**E**, **F**) Knockdown of col-137 by RNAi resulted in a high number of smaller sfGFP::Aβ aggregates with very few bigger sfGFP::Aβ aggregates but no obvious effect on the collagen structure of ROL-6::mScarlet. Scale bars 10μm.

**Author response image 5. sa2fig5:** Knockdown of cuticular collagen dpy-7 affects sfGFP::Aβ aggregates and the ECM morphology represented by collagen ROL-6::mScarlet. Knockdown of dpy-7 by RNAi resulted in the absence of sfGFP::Aβ but also effect on the collagen structure of ROL-6::mScarlet. Scale bars 10μm.

Lastly, we used pharmacological treatments that inhibit ECM remodeling. The rationale was if ECM remodeling is required for sfGFP::Aβ aggregation formation or removal from the cuticle, then these pharmacological interventions should affect sfGFP::Aβ levels. We found that de novo collagen synthesis (BPY as a prolyl 4-hydroxylase inhibitor) reduced sfGFP::Aβ levels, whereas inhibiting collagen crosslinking (BAPN as a lysyl oxidase inhibitor) increased sfGFP::Aβ levels (see new Supporting Figure 4.1). Furthermore, blocking collagen degradation with MMP-inhibitor batimastat resulted in higher sfGFP::Aβ levels (Figure 5B). These data suggest that interfering with ECM remodeling might be functionally implicated in sfGFP::Aβ aggregation.

In summary, we depicted this direct and indirect MoA of ADM-2 as a hypothetical model in Figure 6G. We hope this clarifies the potential MoA of ADM-2.

2) It is unknown whether collagens escalation increases aggregate formation.

We are not sure what the reviewer means by collagen escalation (defined as an increase in the intensity or seriousness of something; an intensification). If collagen escalation refers to collagen dynamics or remodeling, then please see above “indirect MoA of ADM-2” the new pharmacological experiments with collagen biogenesis and stability inhibitors (shown in Supporting Figure 4.1). If escalation refers to higher collagen levels, then we would like to direct the attention to the batimastat experiments in Figure 5A-B. Batimastat is a broad-range metalloproteinase inhibitor, thereby blocking collagen degradation and essentially leading to “overexpression” or collagen intensification in the cuticle. With the batimastat inhibitor, we observed higher sfGFP::Aβ levels (Figure 5B). Whether and how transgenic overexpression of collagen directly would increase Abeta aggregation, we did not investigate for the following reasons. First, there are 180 collagens in *C. elegans* (DOI: 10.1016/j.mbplus.2018.11.001). We would need to generate a “screen” by either transgenic overexpression or using CRISPR activation (DOI:https://doi.org/10.1016/j.jbc.2022.102085). Although this would be feasible, generating that many transgenic lines or sgRNA guides would be another Ph.D. project by itself. Second, we had shown that overexpression of collagen COL-120 increases the levels of other collagens (COL-10, COL-144, etc.) and also enzymes that remodel the ECM (doi: https://doi.org/10.1101/2022.08.30.505802), suggesting the induction of broad ECM dynamics or turnover.

Taken together, this suggests that collagen dynamics or remodeling plays a role in Aβ aggregation levels in the ECM.

3) There is no direct evidence to prove that the GFP-positive aggregates do contain human Aβ. This can be achieved by methods including: Western blot to show aggregated Aβ, electron microscopy to observe Aβ fibrils, and immunostaining showing the colocalization between GFP fluorescence and human Aβ.

To provide evidence that the GFP-positive aggregates contain human Aβ, we performed Western blotting using GFP and human Aβ antibodies. Day 1 wild type (N2) and transgenic LSD2104 sfGFP::Aβ adults were heat-shocked to induce Aβ aggregates. 24 h later, when the remaining sfGFP::Aβ has formed aggregates in the cuticle, animals were harvested for protein isolation. Protein lysates were probed against Aβ [Anti-amyloid β peptide (MOAB-2) pan Antibody, clone 6C3 (Merck #MABN254)], and GFP [Anti-GFP (Roche #11814460001)] from three independent biological trials. Only after heat shock in the transgenic LSD2104 sfGFP::Aβ *C. elegans*, we observed a strong band below 55 kDa and a weaker band below, which correspond to the expected sizes for our transgenic fusion protein of human Abeta (1-42) with super-folder GFP (sfGFP::Aβ) of 44.6 kDa and cleaved one of 42.2 kDa in the anti-Aβ blot. These bands were also observed with the anti-GFP, as shown in Supporting Figure 1.1. Importantly, none of these bands were observed in wild type. This supports the idea that GFP aggregates contain human Aβ in our transgenic strain and that fluorescent sfGFP::Aβ can be used as a proxy for Aβ levels.

We incorporated this into the result section:

“By western blotting using GFP or Aβ antibodies, we confirmed that these sfGFP::Aβ fluorescent puncta contain the human Aβ (Supporting Figure 1.1).”

4) More detailed mechanistic studies, as well as further characterization of the strain, is necessary (e.g., whether this strain shows impaired memory, the key feature of AD-using the well developed chemotaxis assay)

We performed a number of experiments to further characterize the sfGFP::Aβ strain. Unfortunately, our massive efforts resulted in mostly negative data that we present in Author response images 6-11.

**Author response image 6. sa2fig6:** Lifespan assay. At day one of adulthood, wild type (N2) and LSD2104 (GFP_AB) were heat shocked for 2h at 33°C and placed on 50 µM FUdR NGM plates at 20°C to assess their lifespan (Protocol as in Ewald et al., 2016 https://doi.org/10.1111/acel.12509). The GFP_AB (LSD2104 sfGFP::Aβ) strain was shorter-lived as wild type N2. This could be due to the inherent leakiness at 20°C of hsp-16 promoter driving the sfGFP::Aβ expression (as reported for other transgenes and lifespan in Ewald et al., 2016 https://doi.org/10.1111/acel.12509). However, the single heat shock, which induced the sfGFP::Aβ, did not further shorten the lifespan of the transgenic LSD2104 strain. We know that the heat shock had worked due to the slight increase in lifespan of the heat-shock wild type (a phenomenon also reported in Ewald et al., 2016 https://doi.org/10.1111/acel.12509).

**Author response image 7. sa2fig7:** Embryonic survival assay. Upon heat shock, sfGFP::Aβ is expressed in eggs. We aimed to determine the effects of sfGFP::Aβ on embryonic survival and hatching into the first larval stage. 10 N2 or LSD2104 (ss::sfGFP::Aβ) gravid adults were allowed to lay eggs for 3h. Then either heat shocked at 33°C for 3h (HS) or not (No HS). The number of hatched vs non-hatched eggs was scored 24 h later. There was no significant difference under non- or heat-shocked conditions of embryonic survival of LSD2104 vs N2. Note that the y-axis of the graph starts at 80%..

**Author response image 8. sa2fig8:** Development and proteotoxicity stress assay. The rationale of this assay was to determine whether perturbing protein homeostasis via increasing osmolarity would affect larval development. Wild type (N2), GFP-only LSD1091 [Phsp-16.2::ss::GFP] (GFP), and LSD2104 sfGFP::Aβ (GFP AB) adults were heat shocked for 2h at 33°C and F1 eggs were placed on NGM plates containing 0, 50, 100, 200 mM NaCl and the larval stage was scored 48h at 25°C. For N2 only at 200 mM salt, the development was slowed, whereas sfGFP::Aβ (GFP AB) already had a bit slower development, which was further slowed at 200 mM salt. Given that the difference pre-L4 to L4 is a few hours and rather minor, this suggests that sfGFP::Aβ (GFP AB) had minor proteotoxicity effects on development.

**Author response image 9. sa2fig9:** Cuticle integrity assay. Since sfGFP::Aβ forms aggregates in the cuticle (ECM), we sought to determine whether that would affect cuticle barrier function. LSD2104 sfGFP::Aβ adults were heat shocked for 2h at 33°C (HS) or not (no HS) and then placed in low bleach concentration and M9 to determine the time until cuticle breaks. Three biological repeats (1,2,3) are shown that are all non-significant. (Experimental details: 500 µL NaCl + 1mL NaOH (1M) were added to a total volume of 10 mL in M9. A 24-well plate was used to add 1 mL of this bleach solution per well. In each well, 15 worms were placed, at which time point a timer was started. Scored was the time point for each worm cuticle to break by visual observation with a dissecting scope.).

**Author response image 10. sa2fig10:** Touch habituation. Previously, we had shown that overexpression of APL-1 (the homolog of the APP) showed impairment in touch habituation (Ewald et al., 2012, DOI: https://doi.org/10.1523/JNEUROSCI.0495-12.2012). To determine whether sfGFP::Aβ also affects learning plasticity, we compared N2 wild type (WT) with LSD2104 sfGFP::Aβ. Day 1 adults were heat shocked for 3h at 33°C, and 24h later, the responses to touch with a hair lash to the head and tail were recorded. The total time was recorded until the animal became unresponsive (A; data plotted as a bar graph). Also, the total number of touches until the animal became unresponsive was counted, and data was plotted as a bar graph (B). The bar graph shown here represents the cumulative mean data for two independent trials. Error bars are shown as SEM. Each dot represents a single animal. 2-way Anova was performed for statistical analysis. Transgenic LSD2104 sfGFP::Aβ animals had higher resistance towards the touch and took longer time to become habituated to touch as compared to wildtype even under non-heat shocked conditions (i.e., when sfGFP::Aβ was not induced). Heat shock-induced aggregate formation, however, had no effect on touch habituation. This result suggests that Aβ aggregate formation in the extracellular matrix does not affect tactile learning plasticity.

**Author response image 11. sa2fig11:** Chemotaxis assay. Previously, we had shown that overexpression of APL-1 (the homolog of the APP) showed impairment in olfactory and gustatory learning behavior (Ewald et al., 2012, DOI: https://doi.org/10.1523/JNEUROSCI.0495-12.2012). To determine whether sfGFP::Aβ also affects olfactory learning plasticity, we compared N2 wild type (WT) with LSD2104 sfGFP::Aβ. Day-1 adults were heat shocked for 3h at 33°C, and 24h later, a chemotaxis assay was performed. Benzaldehyde was used as the test chemical, while ethanol served as the control. Briefly, a 90 mm unseeded NGM containing petri plate was divided into four quadrants, and a dot was placed equidistant from the center inside each of the quadrants. 1 µL ethanol was spotted in two diagonally opposite quadrants and benzaldehyde in the other two. 1:200 was the dilution used for benzaldehyde (0.005%). 1 µL of 1M Sodium azide was also pipetted in all the four spots. Approx 100-200 animals were exposed to the chemicals by placing them in a 1 cm circle marked in the center. For pre-conditioning, animals were placed in the center and exposed to 0.6 µL of undiluted benzaldehyde by placing the chunks of agar containing the benzaldehyde on the lid of the Petri plate. After pipetting the animals, let the water evaporate before closing the lid. The animals were incubated with benzaldehyde for one hour at 20 ℃. Pre-conditioned animals were assayed for chemotaxis behavior after one hour of incubation with benzaldehyde. After one hour, plates were shifted to 4 ℃ to freeze the animals in their position. One hour later, animals were counted in each quadrant and the center. The Chemotaxis Index (C.I.) was calculated as (the number of animals in test – the number of animals in control) divided by the total number of animals in the plate. Bar graph represents the chemotaxis index as the mean value of three independent trials for each condition. Error bars are SEM. 2-way Anova was performed for statistical analysis. We observed that sfGFP::Aβ transgenic animals had lower C.I. as compared to wild-type, which was not affected by heat-induced aggregate formation. Pre-conditioned animals, as expected, showed reduced C.I. in all conditions. This result suggests that aggregate formation did not affect the chemotaxis behavior towards benzaldehyde in neither naive nor pre-conditioned environments.

Reviewer #1 (Recommendations for the authors):The concerns for the current manuscript are: (1) it remains unclear how ADM-2 removes extracellular aggregates. (2) it is missing whether collagens escalation increases aggregate formation.These two biological processes are critical for understanding the balance in Abeta aggregate formation.

We thank the reviewer for carefully reading our manuscript and identifying these two main concerns. We have addressed them, please see points 1 and 2 above.

Reviewer #3 (Recommendations for the authors):The authors generated a novel sfGFP::Aβ *C. elegans* models of AD that expresses Abeta aggregates extracellularly; using this worm model, they identified that a disintegrin and metalloprotease ADM-2, an ortholog of human ADAM9, participated in removing these extracellular aggregates. This worm model may be very useful to the AD field. Addressing the below questions will improve the quality of the paper.– While the major focus of this paper was on the elimination of Abeta plaques, it is important to give a summary on the multiple causes/risks of AD as elimination of Abeta plaques may not be sufficient to inhibit AD (the recent Abeta antibody by the Swedish company showing 28% reduction of memory loss via an Abeta antibody while significant side effects reported). Pathologies features like Tau tangles, defective mitophagy, senescence, and inflammation should be mentioned (PMID: 23254930; PMID: 30742114; PMID: 30936558 ). The authors may not necessarily have to test whether ADM-2 shows any direct/indirect/non-autonomous effects on Tau pathologies, defective mitophagy, senescence, and inflammation, it could be nice for them to discuss such things in the 'Discussion' section.

We thank the reviewer for taking the time and reading and evaluating our manuscript. We also highly appreciate the reviewer’s idea to determine the effects of ADM-2/ADAM9 on Tau pathology. From a disease modeling perspective, this would be the next step for a 2.0 Alzheimer’s disease model of *C. elegans*. Such a model has been recently established with neuronal expressing human Abeta and humanTau by Benbow et al., 2020 and many of the resulting pathologies were due to the action of Tau (doi: 10.1093/hmg/ddz319). We would expect similar indirect effects of extracellular Abeta in our model with intracellular Tau resulting pathologies. It could also be that the reason for Abeta aggregation in the ECM is to “safe-store” the Abeta and thus prevent toxicity via Tau. This asks for a proper side-by-side comparison and testing of the hypothesis in future work. We hope that providing our extracellular Abeta aggregation in the ECM model will allow many researchers in the field to go on to test their favorite hypotheses.

We did an extensive literature search to find any connection or evidence for ADAM9 to affect Tau pathology. Unfortunately, we were not able to find any evidence in the literature. Thus, this idea needs to be experimentally tested. We did not include this idea in the discussion since it is a completely new project, and evidence is lacking to make such a direct leap from the literature.

– Any evidence to validate that the sfGFP::Aβ strain expresses Abeta1-42, rather than Abeta3-42 (as happened in other Abeta strains before)?

We have not obtained any evidence to validate this for the following reasons that it seems unlikely to be the case. To experimentally prove this, we would need to run surface-enhanced laser desorption ionization-time of flight mass spectrometry like McColl et al., 2009 did (DOI: 10.1074/jbc.C109.028514). As McColl and colleagues have stated: “The likely cause is aberrant cleavage of the N-terminal signal peptide incorporated in the *C. elegans* transgene (Figure 1D).” (DOI: 10.1074/jbc.C109.028514). Given this, it seems unlikely that the first 3 amino acids would be missing since we, for this reason, preceded the super folder GFP the Aβ1-42 sequence.

We emphasized this in the text.

“A spacer sequence was placed between the sfGFP and the Aβ to allow the comparably smaller-sized Aβ to move and interact freely to form aggregates (Figure 1A, 1B). The full length of the Aβ1-42 peptide is essential for its aggregation (Mccoll et al., 2012). In our construct, Aβ is preceded by the sfGFP and spacer sequence, which prevents the truncation of the first few amino acids observed in many previous *C. elegans* Aβ models (McColl et al., 2009).”

– Did the single heat shock induction (33 C for 2 h) induce signalling pathways that may contribute to 'artificial effects' in Abeta clearance?

We assume that the single heat shock will induce small heat shock proteins, which peak after 8-12 hours and remain for at least 32 hours. We had previously characterized this heat shock response time course using TJ375 [Phsp-16.2::GFP] in microfluidics and western blotting with HSP-16.2 antibody (Figure 1 and Sup. Figure 1 in Verttti-Quintero et al., 2021 DOI: 10.1002/smll.202102145).

These small heat shock proteins probably help clear intracellular sfGFP::Aβ within the first 5 hours in many tissues, as we observed and illustrated in Figure 1D.

As we focus on extracellular Abeta aggregation, it is unknown whether secreted HSP90 or similar proteins would participate in extracellular Abeta aggregation or removal. In our screen, where we examined the Abeta aggregation in the ECM (cuticle), non of the heat-shock response family member showed a significant impact on cuticular-localized sfGFP::Aβ fluorescent levels.

We had knockdown hsp-1, hsp-70, hsp-4, hsp-90, hsp-110, hsp-12.3, hsp-12.6, hsp-60, hsp-75, hsp-6, hsp-16.2, and hsf-1 (Supporting File 1). Only knockdown of hsp-110/HSPA4,1 resulted in higher sfGFP::Aβ fluorescent levels (Figure 3C), suggesting minor involvement of the heat shock response pathway. However, we had not tested whether hsp-110 affects sfGFP::Aβ aggregation or removal. Thus, although we can not formally exclude it, it seems that the heat shock response pathway has a minor or no role in sfGFP::Aβ removal.

– As AD pathologies are primarily in the brain region, a good worm model should also mimic this feature to enable a higher likelihood of data interpretation from worms to humans. As the 'nerve ring' is like the 'brain', have the authors considered expressing Abeta plaques (primarily) in this region?

We chose hsp-16.2 promoter since it is inducible (by passing development), and the hsp-16.2 promoter drives expression in many tissues but predominantly in neurons and hypodermis (Bacaj and Shaham, 2007). We detected some neuronal cell bodies that could be in the nerve ring that might express sfGFP::Aβ (Author response image 12).

**Author response image 12. sa2fig12:** Head region of sfGFP::Aβ and collagen type IV/emb-9::mCherry. We performed confocal imaging of the head region of sfGFP::Aβ and collagen type IV/emb-9::mCherry. By the bulb outlined by the basement membrane of emb-9::mCherry (in red), there are several cells in green (i.e., sfGFP::Aβ) that could be the nerve ring and neurons.

However, to directly address this, we generated a new transgenic strain using the pan-neuronal rab-3 reporter.

[Prab-3::ssSel1::FLAG::superfolderGFP::spacer::humanAmyloidBeta1-42::let-858-3’UTR; pRF4 rol-6(su1006)]. Unfortunately, these transgenic *C. elegans* arrest in L1 (Author response image13). Thus, more future research is required to establish a non-inducible neuronal expression sfGFP::Aβ strain.

**Author response image 13. sa2fig13:** Pan-neuronal expressing sfGFP::Aβ *C. elegans* arrest in L1. Shown is a GFP channel image above and the same image in the bright field below. GFP-positive animals are arrested in L1 (green tiny worms), and all the adult worms visible in the bright field image are GFP-negative, suggesting that neuronal expressing sfGFP::Aβ *C. elegans* arrest in L1.

– The authors noticed an important phenomenon that 'different collagens to ameliorate or potentiate Aβ aggregation': what are the underlying molecular mechanisms? It could be nice to discuss underlying mechanisms in the 'Discussion section'.

To address this, we expanded, validating more collagen hits from our screen. As shown above, in point 1 and Author response images 4 and 5, we show different sfGFP::Aβ aggregation patterns or levels by knocking down these collagen hits.

Based on their sequence, they fall into different cuticular collagen classes (see classification and tool ((http://ce-matrisome-annotator.permalink.cc/ and *C. elegans* collagen database, CeColDB, available at: http://CeColDB.permalink.cc/ in Teuscher et al., 2019 https://doi.org/10.1016/j.mbplus.2018.11.001). Thus, it is hard to predict their functional role or even to speculate on the underlying mechanism besides inducing or altering collagen remodeling.